# The role of ATP synthase subunit e (ATP5I) in mediating the metabolic and antiproliferative effects of metformin in cancer cells

Guillaume Lefrançois[1,2], Emilie Lavallée[3], Marie-Camille Rowell[2], Véronique Bourdeau[2], Farzaneh Mohebali[1], Thierry Bertomeu[4], Ana Maria Duman[2], Maya Nikolova[2], Mike Tyers[4], Simon-Pierre Gravel[3,4]*, Andreea R Schmitzer[1]*, Gerardo Ferbeyre[2]*

[1]Department of Chemistry, University of Montreal, Montreal, Canada; [2]Department of Biochemistry and Molecular Medicine, CR-CHUM and Montreal Cancer Institute, University of Montreal, Montreal, Canada; [3]Faculté de Pharmacie, University of Montreal, Montreal, Canada; [4]Institute for Research in Immunology and Cancer, University of Montreal, Montreal, Canada

*For correspondence:
sp.gravel@umontreal.ca (S-PG);
ar.schmitzer@umontreal.ca
(ARS);
g.ferbeyre@umontreal.ca (GF)

**Competing interest:** The authors declare that no competing interests exist.

## eLife Assessment

This **valuable** manuscript describes ATP5I, a subunit of F1Fo-ATP synthase, as a key target of medicinal biguanides. The knockout of ATP5I in pancreatic cancer cells mimics biguanide treatment, inducing a metabolic switch from OXPHOS to glycolysis due to a compromised expression of the Complex I protein NDUFB8. This results in a markedly decreased NAD/NADH ratio and decreased cell proliferation. These **solid** findings point out ATP5I as a promising mitochondrial target for cancer therapies and contribute to our understanding of metformin's mechanism of action since many of its molecular mechanisms remain poorly understood.

**Abstract** Here, we identify the subunit e of $F_1F_0$-ATP synthase (ATP5I) as a target of metformin, a first-in-class antidiabetic biguanide. ATP5I maintains the stability of $F_1F_0$-ATP synthase dimers, which is crucial for shaping cristae morphology. We demonstrate that ATP5I interacts with a biguanide analogue in vitro, and disabling its expression by CRISPR–Cas9 in pancreatic cancer cells leads to the same phenotype as biguanide-treated cells, including mitochondrial morphology alterations, reduction of the $NAD^+$/NADH ratio, inhibition of oxidative phosphorylation (OXPHOS), rescue of respiration by uncouplers, and a compensatory increase in glycolysis. Notably, metformin disrupts $F_1F_0$-ATP synthase oligomerization, leading to the accumulation of vestigial assembly intermediates in pancreatic and osteosarcoma cancer cells, a phenotype also observed upon ATP5I inactivation in pancreatic cancer cells. Moreover, ATP5I knockout (KO) cells exhibit resistance to the antiproliferative effects of biguanides, but reintroduction of ATP5I rescues the metabolic and antiproliferative effects of metformin and phenformin. Finally, a genome-wide CRISPR screening in NALM-6 lymphoma cells revealed that metformin-treated cells exhibit genetic interaction profiles similar to those observed with the $F_1F_0$-ATP synthase inhibitor oligomycin, but not with the complex I inhibitor rotenone. This provides unbiased support for the relevance of the newly proposed target.

## Introduction

Discovering safe and effective therapeutic targets for cancer treatment remains a major challenge in biomedical research (*Emmerich et al., 2021*). Recently, targeted therapies against complex I of the respiratory chain have garnered considerable attention, offering a promising strategy for cancer treatment due to their potential to disrupt the energy metabolism of tumor cells (*Janku et al., 2021*; *Yap et al., 2023*; *Ashton et al., 2018*; *Vasan et al., 2020*). However, despite initial promises, clinical trials with potent complex I inhibitors have been compromised by severe toxicities, raising concerns about their clinical viability. Given the critical role of mitochondria in cancer cells, there is an urgent need for more effective and safer alternatives to target them (*Zhang and Dang, 2023*; *Machado et al., 2023*). In this context, medicinal biguanides such as metformin and the more lipophilic phenformin are emerging as attractive alternatives. These biguanides are considered moderate respiratory chain inhibitors and offer a potential avenue for targeting mitochondrial metabolism in cancer (*Xu et al., 2020*).

Metformin has been used since the 1950s for its anti-hyperglycemic properties (*Bailey, 2017*). As such, it has a well-established safety profile and is commonly used for the treatment of type II diabetes (*WHO, 2019*). Several epidemiological studies showed that prolonged use of metformin in diabetic patients reduced the incidence of several cancers (*Evans et al., 2005*; *Pernicova and Korbonits, 2014*), particularly pancreatic cancer (*Hu et al., 2023*; *Gong et al., 2014*). However, its clinical use has proven ineffective in patients with advanced pancreatic cancer (*Kordes et al., 2015*), suggesting that its antitumoral mechanism is more preventive than therapeutic (*Quinn et al., 2013*; *Pollak, 2012*). The more potent phenformin (*Bailey, 2017*), which was withdrawn from the market in the 1970s for its anti-hyperglycemic applications due to deaths attributed partly to lactic acidosis (*Kwong and Brubacher, 1998*), is currently undergoing clinical trials in cancer patients (*García Rubiño et al., 2019*).

Unlike other inhibitors of the respiratory chain, the mechanism of action of medicinal biguanides remains elusive (*Viollet et al., 2012*; *Foretz et al., 2019*). However, compelling evidence indicates that energy metabolism plays a pivotal role in their effectiveness (*Andrzejewski et al., 2014*; *Deschênes-Simard et al., 2019*). At supra-pharmacological doses (1–10 mM), typically 100–1000 times higher doses than those used for anti-hyperglycemic effects, metformin moderately inhibits the activity of complex I of the respiratory chain in several cellular models (*Wheaton et al., 2014*; *Bridges et al., 2014*; *ElMir et al., 2000*). This inhibition subsequently decreases OXPHOS activity, leading to reduced cellular respiration and energy production. In response, cells adapt by shifting metabolism to glycolysis and decreasing their anabolic activity through the activation of the energy sensor AMPK (*Viollet et al., 2012*; *Andrzejewski et al., 2014*; *Wheaton et al., 2014*). Nonetheless, to observe a decrease in isolated complex I activity in in vitro models, metformin doses of up to 50 mM are required (*Bridges et al., 2014*; *ElMir et al., 2000*; *Bridges et al., 2023*). A recent cryo-electron microscopy study revealed that a metformin analogue binds to complex I in a specific conformation with a binding site unique to this analogue (*Bridges et al., 2023*). However, further work using in vivo models is necessary to validate the biological meaning of this interaction as a critical site of action of metformin (*Bridges et al., 2023*; *Matsuzaki and Humphries, 2015*).

Additional targets such as glycerol phosphate dehydrogenase (*Madiraju et al., 2014*), complex IV (*LaMoia et al., 2022*), PEN2 (*Ma et al., 2022*), and $F_1F_0$-ATP synthase (*Bridges et al., 2014*) have also been suggested for metformin. Metformin may interact with various targets in a nonspecific manner (*Monera et al., 1994*; *Mayo and Baldwin, 1993*; *Batchelor et al., 2004*; *England and Haran, 2011*) because of its structural similarity with guanidine, a potent chaotropic agent. These various targets potentially exert pleiotropic effects (*Viollet et al., 2012*; *Foretz et al., 2019*) that could act synergistically.

It has been shown that atrazine, a molecule derived from biguanides, binds and inhibits $F_1F_0$-ATP synthase (*Hase et al., 2008*). Additionally, a study from our group revealed that medicinal biguanides likely disrupt cristae organization (*Hébert et al., 2021*), a crucial process mediated by the oligomerization of $F_1F_0$-ATP synthase and resulting in the formation of 'onion-like structures' (*Paumard et al., 2002*; *Habersetzer et al., 2013b*; *Habersetzer et al., 2013a*; *Zick et al., 2009*). It is proposed that complex I and $F_1F_0$-ATP synthase of the respiratory chain may be particularly sensitive to biguanide inhibition due to the conformational mobility of their catalytic interfaces (*Bridges et al., 2014*). Disrupting the folding of the mitochondrial inner membrane could alter the organization and functions

of other respiratory chain complexes (*Davies et al., 2011*). It remains an open question whether biguanides act through multiple targets or a single primary target.

Here, we identify another target of biguanides: the e subunit of $F_1F_0$-ATP synthase (ATP5I), a transmembrane protein involved in the dimerization and assembly of this supramolecular complex (*Paumard et al., 2002*; *Habersetzer et al., 2013b*; *Habersetzer et al., 2013a*). Although medicinal biguanides are known to accumulate in mitochondria and disrupt the respiratory chain by inhibiting complex I, no direct evidence of their interaction with a subunit of $F_1F_0$-ATP synthase has been reported until now. CRISPR–Cas9-mediated KO of ATP5I recapitulates many of the effects of metformin in pancreatic cancer cells and reduces metformin sensitivity. Conversely, treatment with metformin interferes with the oligomerization and assembly of the $F_1F_0$-ATP synthase complex as happens in cells without ATP5I. Interestingly, the genetic signature of cells treated with metformin is more similar to that of cells treated with an $F_1F_0$-ATP synthase inhibitor than to a complex I inhibitor. Taken together, this work adds the $F_1F_0$-ATP synthase and its subunit ATP5I as a bona fide target of biguanides.

## Results

### Identification of ATP synthase subunit e (ATP5I) as a mitochondrial biguanide binding protein

To identify proteins that bind biguanides, we synthesized a biotin functionalized biguanide or BFB (*Figures 1A, B*) to perform an affinity-based pull-down assay with streptavidin beads (*Sato et al., 2010*). We first assessed the biological activity of BFB comparing it to metformin in the ability to activate AMPK and inhibit cell proliferation in KP-4 pancreatic cancer cells. BFB activated AMPK and its downstream target ACC as indicated by their increased phosphorylation states (*Figure 1B*) and inhibited KP-4 cell growth with an $EC_{50}$ of 1.0 ± 0.2 mM similar to metformin (*Figure 1C*). Additionally, immunofluorescence experiments with fluorophore-conjugated streptavidin confirmed that BFB accumulates in mitochondria, as shown by colocalization with the mitochondrial protein TOMM20 (*Figure 1D, E*).

After confirming that BFB exhibited activity comparable to that of metformin, we performed pull-down assays using streptavidin immobilized on sepharose beads on mitochondrial-enriched cell extracts to analyze interacting proteins by mass spectrometry. Given the significant issue of nonspecific binding associated with this technique, we performed multiple parallel pull-down experiments. These included a control biotin-conjugated amine derivative (BFA) possessing the same linker as BFB (*Figure 1—figure supplement 1C*). Mass spectrometry analysis results identified a total of 69 proteins. Among these, 30 proteins showed specific interaction with BFB under the elution condition with metformin (*Figure 1—figure supplement 2*). Keratins identified in this way were considered contaminants. However, two mitochondrial proteins stood out, arginase isoform 2 and ATP synthase subunit e (ATP5I). Arginase binds arginine which has a guanidinium group structurally related to metformin (*Caldwell et al., 2018*), while ATP5I is part of the peripheral stalk of the $F_1F_0$-ATP synthase (*Pinke et al., 2020*).

### Biguanide pharmacophore interacts specifically with ATP5I

As ATP5I was not previously shown to bind biguanides, we first confirmed this interaction through immunoblot analysis in independent pull-down experiments. The results indicate that BFB allows specific interaction of ATP5I in streptavidin pull-down while BFA failed to do so. Of note, we observed a nonspecific band (≈11 kDA) in all pull-down conditions with streptavidin beads, possibly associated with the denatured streptavidin monomer in heated SDS buffer (*Figure 1F*). We then used surface plasmon resonance (SPR) to characterize the binding of BFB and purified recombinant ATP5I. Our purification steps yielded a highly pure recombinant protein, mainly organized as oligomers (mostly decamers) in solution (*Figure 1—figure supplement 3A–C*). Although SPR is sensitive (*Masson et al., 2006*; *Sun et al., 2023*), the size difference between BFB and recombinant ATP5I prompted us to immobilize the small molecule BFB on the gold surface. Adding increasing concentrations of recombinant ATP5I allowed us to estimate an affinity constant (KD ≈ 11.10 μM; Rmax ≈ 39.40 RU) significantly lower than typical metformin concentrations that affect complex I in the mM range (*Figure 1G, H*).

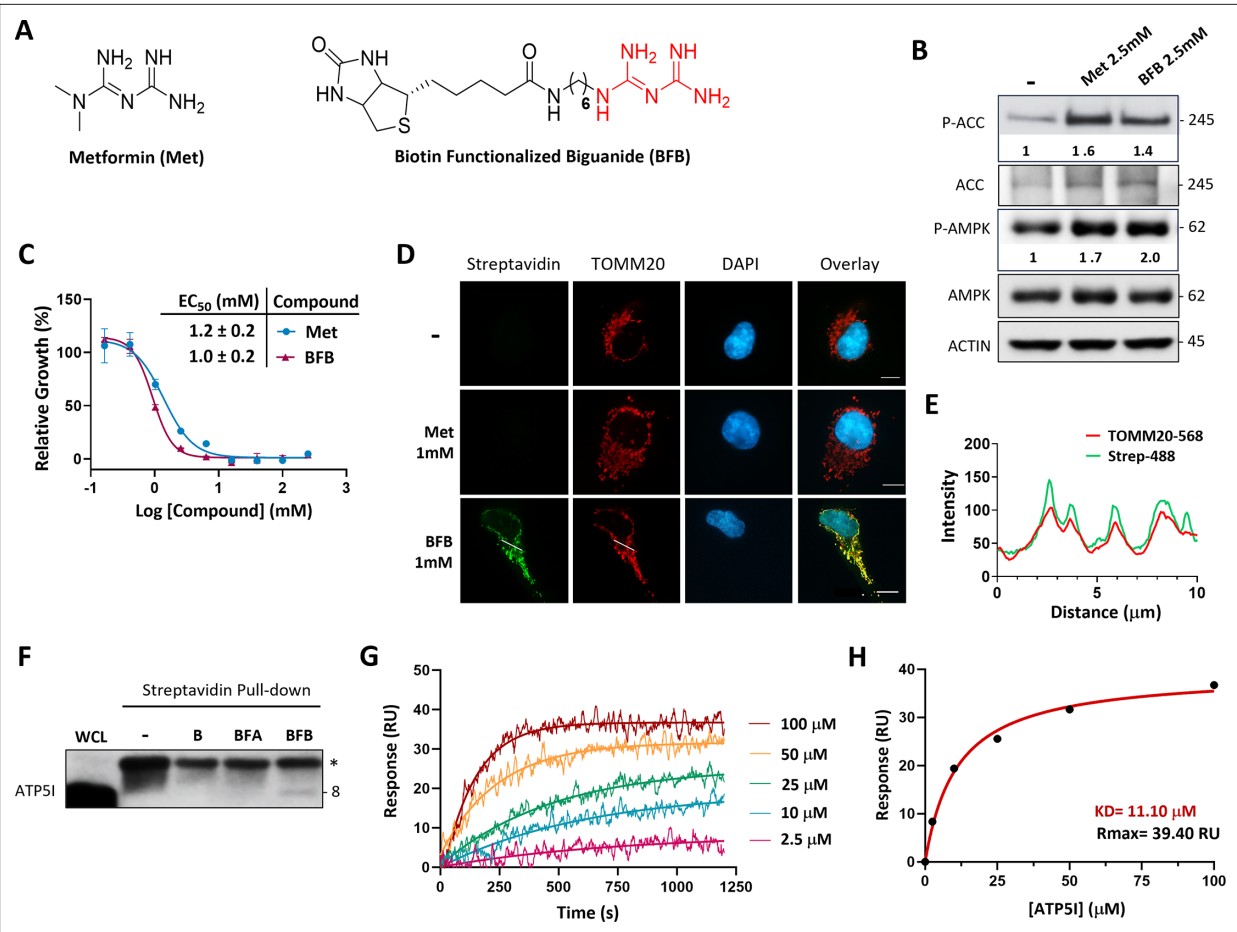

**Figure 1.** Biguanide pharmacophore interacts with ATP synthase subunit e (ATP5I). (**A**) Design of bio-inspired probe biotin functionalized biguanide (BFB) based on the structure of metformin (Met). (**B**) Immunoblots for the phosphorylation of AMPK (Thr172) and ACC (Ser79) in extracts from KP-4 pancreatic cancer cells treated with 2.5 mM Met or BFB for 16 hr. β-ACTIN was used as loading control. (**C**) Representative quantification of cell viability and growth with corresponding $EC_{50}$ values of 3-day treatments with metformin (Met) or BFB in KP-4 cells. Values represent the mean ± standard deviation of $N = 3$. (**D**) Representative images of mitochondria and BFB localization in cells as in (**B**). Cells were treated with 1 mM of metformin (Met) or BFB for 16 hr and mitochondrial signal and BFB localization were analyzed by co-immunofluorescence using streptavidin fluorophore conjugate and anti-TOMM20 antibody, scale bar = 10 μm. Cells untreated (-) and treated with 1 mM Met were used as negative controls. (**E**) Colocalization between TOMM20 (TOMM20-568) and streptavidin (Strep-488) fluorophores was analyzed for the BFB condition from (D) through job plot intensity profile. (**F**) Pull-down validation experiments with streptavidin beads alone (-), D-biotin (B), biotin functionalized amine (BFA) and BFB using antibody followed by immunoblot against ATP5I in cells as in in extracts from HEK-293T embryonic kidney cells. The whole cell lysate (WCL) was added as control. (**G**) Binding interactions studies of BFB with recombinant purified ATP5I (rATP5I) using Surface Plasmons of Resonance (SPR). Representative sensorgrams show affinity kinetics of BFB and rATP5I. BFB was exposed onto streptavidin immobilized sensor chip and several concentrations of rATP5I were added until saturation of the signal. RU: Resonance Units. (**H**) Binding affinity curve obtained from each steady state from (G). $K_D$ refers to the dissociation equilibrium constant and Rmax represent the theoretical maximum response.

The online version of this article includes the following source data and figure supplement(s) for figure 1:

**Source data 1.** Contain details of chemical synthesis and structural characterization of the compounds made for this article.

**Source data 2.** PDF file containing original western blots for *Figure 1B, F*.

**Source data 3.** Contain original TIF files used to make *Figure 1B, F*.

**Figure supplement 1.** Synthetic routes for biotin-NHS, biotin-functionalized biguanide, and biotin-functionalized amine probes.

**Figure supplement 2.** Proteins were isolated from streptavidin-coated beads after affinity purification using the biotinylated biguanide probe (BFB), followed by competitive elution with metformin (50 mM).

**Figure supplement 3.** Purification and size-exclusion chromatography characterization of recombinant ATP5I.

## ATP5I knockout in pancreatic cancer cells alters the organization of mitochondrial networks

In yeast, inhibition of the subunit e's equivalent impaired mitochondrial inner membrane folding (*Arnold et al., 1997*). This structural deficiency in yeast correlates with decreased dimerization of $F_1F_0$-ATP synthase, highlighting the essential role of ATP5I in dimer stability. However, yeast's subunit e is not essential for the catalytic activity of $F_1F_0$-ATP synthase (*Arnold et al., 1997*; *Arselin et al., 2003*; *Brunner et al., 2002*; *Arselin et al., 2004*; *Everard-Gigot et al., 2005*). In human cells, ATP5I is also crucial for dimer stability (*Habersetzer et al., 2013a*) but lacks thorough characterization regarding its roles in cellular energy metabolism. To elucidate its function in cancer and its relevance to cells treated with medicinal biguanides, we developed CRISPR-based reagents to inhibit ATP5I expression in KP-4 pancreatic cancer cells.

Initially, two clones each of the two guides (sgATP5I #1 and sgATP5I #2) were isolated, and their mitochondrial phenotype characterized. First, we measured levels of several proteins from the $F_1F_0$-ATP synthase and respiratory complexes. The results show that all ATP5I knockout clones (ATP5I KO) did not express ATP5I or its partner ATP5L (subunit g in yeast) and had lower levels of ATP5O (oligomycin sensitivity-conferring protein, OSCP in yeast) (*Figure 2A*). This result is consistent with previous data in yeast where knockout of subunit e also led to a decrease in subunit g (*Everard-Gigot et al., 2005*). ATP5I KO cells also exhibited decreased expression of complex I NDUFB8 protein and complex IV COX II protein compared to control clones expressing a guide against GFP (sgGFP) (*Figure 2A*, *Figure 2—figure supplement 1A*). A moderate to no reduction in mRNA levels encoding these proteins was observed (*Figure 2—figure supplement 1B*) and cannot account for the down-regulations at the protein level. Additionally, the mitochondrial/nuclear DNA ratio (*Figure 2—figure supplement 1C*) indicates no reduction in mitochondrial number, suggesting ATP5I may play a role in maintaining the stability of certain subunits of complexes I and IV. Consistent with this finding, the absence of subunit e in yeast also controls the stability of $F_1F_0$-ATPase proteins subunit g and subunit k (*Everard-Gigot et al., 2005*).

We then examined mitochondrial morphologies using fluorescence microscopy in ATP5I knockout (KO) cells, revealing a disruption in the mitochondrial network characterized by predominantly punctate mitochondria (*Figure 2B–F*). Relative quantification showed that about ATP5I KO cells exhibited a punctate phenotype, characterized by fewer branches and branch endpoints and reduced diameter (*Figure 2C–F*). These findings confirm that ATP5I plays a crucial role in the organization of the mitochondrial network in KP-4 cells. Treatment of the same cell line with medicinal biguanides resulted in similar alterations in mitochondrial network organization (*Hébert et al., 2021*), reinforcing the hypothesis that ATP5I may be a target of biguanides.

Finally, since ATP5I controls the dimerization and assembly of the $F_1F_0$-ATP synthase, we investigated whether metformin treatment would affect this process. Supporting this model, BN-PAGE analysis of KP-4 cells treated with metformin (10 mM for 3 days) reveals a decrease in oligomeric forms and a corresponding accumulation of intermediate vestigial complexes of lower molecular weight than the monomeric enzyme, as described in cells having CRISPR-mediated inactivation of ATP5I (*He et al., 2018*; *Figures 2*). Notably, this effect is not observed after short-term exposure (16 hr), suggesting that the drug may inhibit the assembly of the $F_1F_0$-ATP synthase but does not disrupt already formed complexes. This increase in assembly intermediates of the $F_1F_0$-ATP synthase upon metformin treatment mimics the phenotype of ATP5I knockout cells, where, in addition, the dimeric and monomeric forms of the enzyme are also scarce (*Figure 2G*). To broaden the significance of this finding, we reproduced it in U2OS cells (*Figure 2G*). Finally, rotenone, a complex I inhibitor, does not induce the formation of these vestigial complexes, indicating that the effect is not secondary to complex I alterations or reduced respiration induced by metformin.

## ATP5I knockout desensitizes pancreatic cancer cells to biguanides

Given the uncertain role of ATP5I in cellular energy metabolism, we investigated the effects of ATP5I knockout (KO) on mitochondrial bioenergetics in KP-4 cells. The results reveal that ATP5I deletion significantly decreases the $NAD^+/NADH$ ratio (*Figure 3A*), most significantly affecting $NAD^+$ concentration (*Figure 3A*, *Figure 3—figure supplement 1A, B*). Our results also indicate a decrease in the oxygen consumption rate (OCR) ove extracellular acidification rate (ECAR) ratio (*Figure 3B*, *Figure 3—figure supplement 1C, D*), suggesting a reduced respiration associated with a compensatory increase

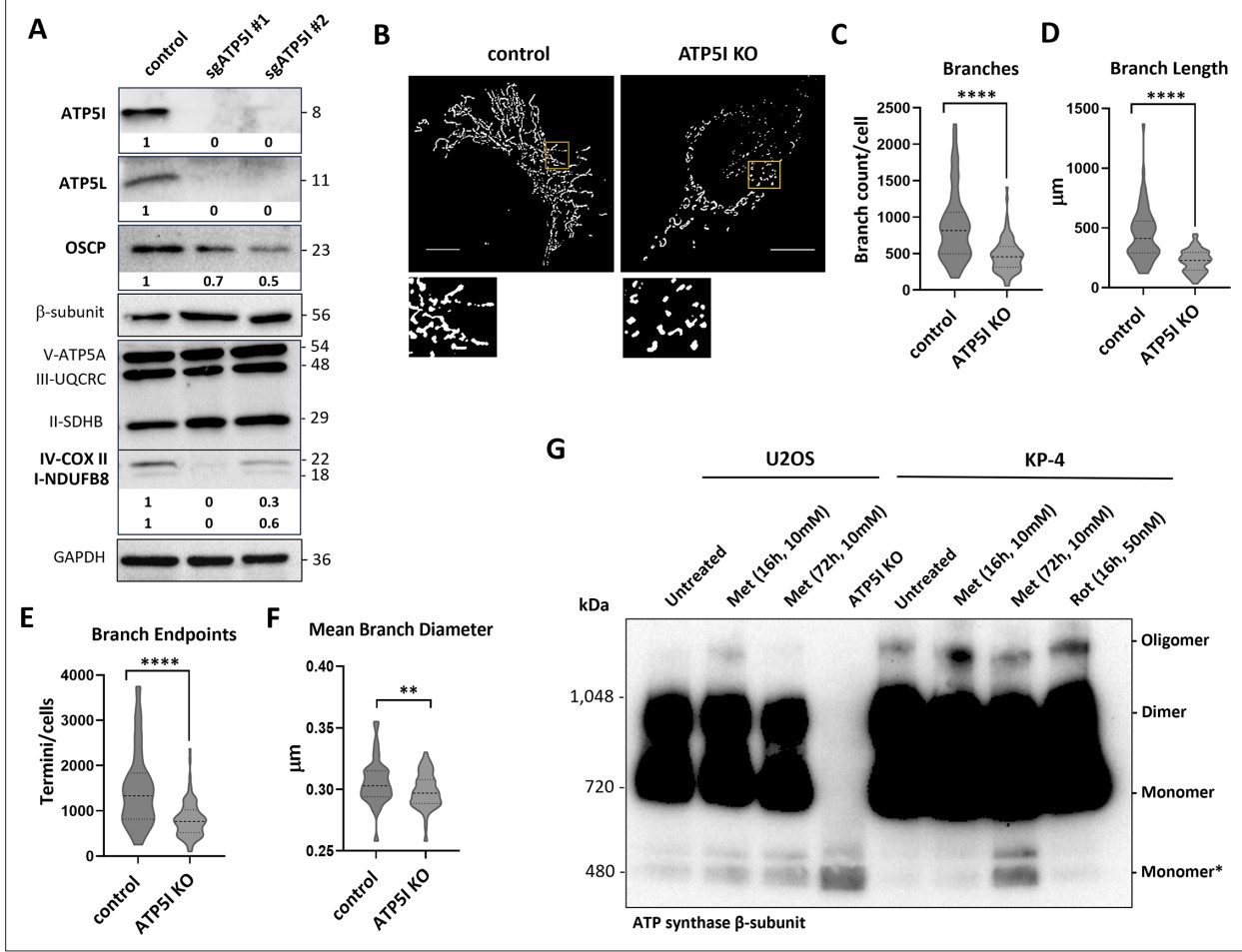

**Figure 2.** ATP5I knockout in pancreatic cancer cells alters the organization of the mitochondrial network. (**A**) Immunoblot for the indicated proteins in extracts from clones of KP-4 cells expressing a control small guide RNA against GFP (sgGFP: control) or two different sgRNAs against ATP5I (sgATP5I #1 and sgATP5I #2). GAPDH antibody was used as a loading control. (**B**) Representative images of mitochondrial morphologies visualized by TOMM20 immunofluorescence (scale bar = 10 μm). A magnified inset (yellow box) is shown for each image to highlight mitochondrial structural details. All images were analyzed using the Mitochondria Analyzer plugin in Fiji (ImageJ). Quantitative analysis of key mitochondrial parameters: (**C**) number of branches, (**D**) total branch length, (**E**) number of branch endpoints, and (**F**) mean branch diameter. Data represent mean ± standard deviation from $N$ = 3 independent clones, with 50–100 cells analyzed per clone. ns: not significant, **p < 0.01, and ****p < 0.0001 using an unpaired Student's $t$-test. (**G**) Representative Blue Native-PAGE followed by western blotting using an antibody against the β-subunit of the $F_1$ domain of ATP synthase in KP-4 or U2OS cells treated with metformin (10 mM, 16 hr or 3 days), or in ATP5I knockout cells (ATP5I KO). Monomer* indicates the assembly intermediates of the $F_1F_0$-ATP synthase known to accumulate after disabling ATP5I.

The online version of this article includes the following source data and figure supplement(s) for figure 2:

**Source data 1.** PDF file containing original western blots for *Figure 2A, G*.

**Source data 2.** Contain original TIF files used to make *Figure 2A, G*.

**Figure supplement 1.** Loss of ATP5I disrupts OXPHOS complex protein expression in KP-4 cells.

**Figure supplement 1—source data 1.** PDF file containing original western blot for *Figure 2—figure supplement 1*.

**Figure supplement 1—source data 2.** Contain original TIF files used to make *Figure 2—figure supplement 1*.

**Figure supplement 2.** Quantification of immunoblot in *Figure 2G*.

**Figure supplement 2—source data 1.** PDF file containing original western blot for *Figure 2—figure supplement 2*.

**Figure supplement 2—source data 2.** Contain original TIF files used to make *Figure 2—figure supplement 2*.

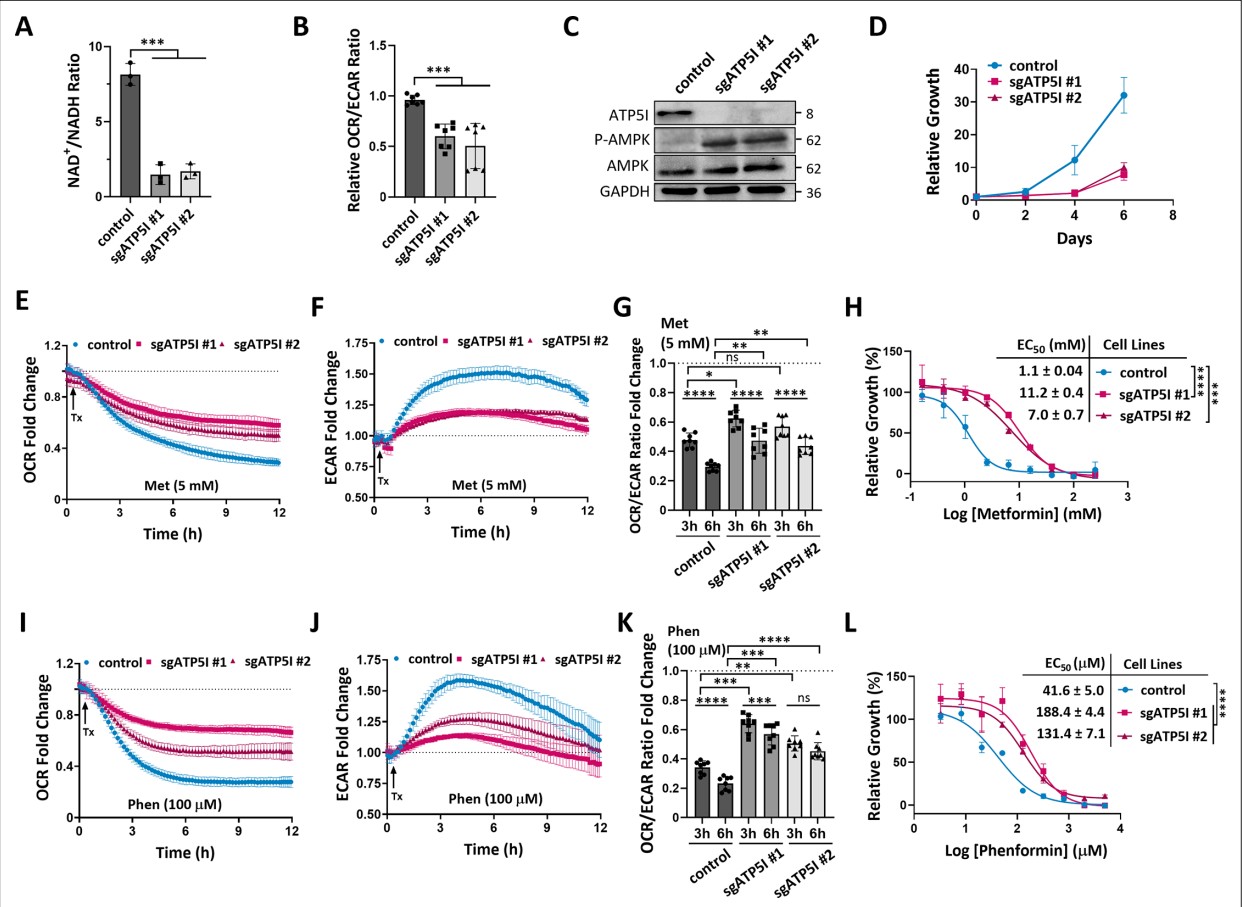

**Figure 3.** ATP5I knockout desensitizes pancreatic cancer cells to biguanides. (**A**) Quantification of NAD⁺/NADH ratio in KP-4 cells expressing a control small guide RNA against GFP (control) or a representative clone of two different guides targeting ATP5I (sgATP5I #1 or sgATP5I #2). Values represent the mean ± standard deviation of $N = 3$. ***$p < 0.001$ using an unpaired Student's $t$-test. (**B**) Relative quantification of oxygen consumption rate (OCR) over extracellular acidification rate (ECAR) by Seahorse analysis in cells as in (A). Values represent the mean ± standard deviation of at least $N = 3$. ***$p < 0.001$ using a paired Student's $t$-test. (**C**) Immunoblot for total and phosphorylated levels of AMPK (Thr172) protein in extracts from cells as in (A). ATP5I confirms loss of expression in KO, and GAPDH was used as loading control. (**D**) Growth curves of cells as in (A) measuring the relative number of cells over 6 days. Media was changed every 2 days. (**E**) Representative kinetic curves of OCR in cells as in (**A**) treated with 5 mM of metformin (Met) relative to control treated cells using Seahorse. (**F**) Representative kinetic curves of ECAR in cells as in (A) treated with 5 mM metformin (Met) relative to control treated cells (dashed line) using Seahorse. (**G**) Quantification of OCR/ECAR ratio fold change at 3 and at 6 hr from kinetic curves (**E, F**). Values represent the mean ± standard deviation of $N = 3$. ns: not significant, *$p < 0.05$, **$p < 0.01$, ****$p < 0.0001$ using a repeated measures (RM) one-way ANOVA with Sidak's multiple comparison test. (**H**) Representative growth of cells as in (A) exposed to different concentrations of metformin for 3 days with corresponding EC₅₀ values of metformin. Values represent the mean ± standard deviation of $N = 3$. ***$p < 0.001$ and ****$p < 0.0001$ using an unpaired Student's $t$-test. (**I**) Representative kinetic curves of OCR in cells as in (A) treated with 100 µM phenformin (Phen) relative to control treated cells using Seahorse. (**J**) Representative kinetic curves of ECAR in cells as in (A) treated with 100 µM of phenformin (Phen) relative to control treated cells (dashed line) using Seahorse. (**K**) Quantification of OCR/ECAR ratio fold change at 3 and at 6 hr from kinetic curves (**I, J**). Values represent the mean ± standard deviation of at least three biological replicates. ns: not significant, **$p < 0.01$, ***$p < 0.001$, ****$p < 0.0001$ using an RM one-way ANOVA with Sidak's multiple comparison test. (**L**) Representative growth of cells as in (A) exposed to different concentrations of phenformin for 3 days with corresponding EC₅₀ values. Values represent the mean ± standard deviation of $N = 3$. ****$p < 0.0001$ using an unpaired Student's $t$-test.

The online version of this article includes the following source data and figure supplement(s) for figure 3:

**Source data 1.** PDF file containing original western blots for *Figure 3*.

**Source data 2.** Contain original TIF files used to make *Figure 3*.

**Figure supplement 1.** ATP5I loss disrupts NAD metabolism, mitochondrial respiration, glycolytic dependence, and metformin sensitivity in KP-4 cells.

**Figure supplement 1—source data 1.** PDF file containing original western blots for *Figure 3—figure supplement 1*.

**Figure supplement 1—source data 2.** Contain original TIF files used to make *Figure 3—figure supplement 1*.

in glycolysis. This metabolic reorganization in ATP5I KO cells renders them up to 6 times more sensitive to glycolysis inhibition with 2-D-deoxyglucose (*Figure 3—figure supplement 1E*), similar to control cells treated with 2.5–5 mM metformin (*Figure 3—figure supplement 1F*). These findings confirm that both ATP5I deletion and metformin treatment disrupt respiration, conferring a dependency on glycolysis (*Ben Sahra et al., 2010*).

Moreover, the inhibition of the mitochondrial respiratory chain observed in ATP5I KO cells induces AMPK activation (*Figure 3C*), similarly to control cells treated with metformin (*Figure 3—figure supplement 1G*). However, the increased phosphorylation state of AMPK in ATP5I KO cells treated with metformin at these concentrations suggests that metformin can still activate AMPK via interactions with other targets such as PEN2 (*Ma et al., 2022*).

Energy stress resulting from ATP5I KO in KP-4 cells leads to a noticeable slowdown in cell growth (*Figure 3D*), although this effect can be reversed by supplementing the cell culture medium with pyruvate and uridine (*Figure 3—figure supplement 1H*). These findings suggest that while ATP5I is not essential for growth in KP-4 cells, its deletion subtly affects energy metabolism, akin to the effects seen in KP-4 cells treated with medicinal biguanides (*Hébert et al., 2021*).

Seahorse kinetic monitoring of control and ATP5I KO cells treated with 5 mM metformin reveals, as expected, that metformin causes a decrease in oxygen consumption (*Figure 3E*) and an increase in glycolysis (*Figure 3F*) over time. However, these effects are more pronounced in control cells than what is observed in ATP5I cells, particularly in terms of glycolysis activation. The OCR/ECAR ratio also decreases more in control cells upon metformin treatment than in ATP5I KO cells (*Figure 3G*), suggesting reduced sensitivity to metformin in the KO cells. This resistance is also evident in the relative growth of cells under increasing doses of metformin. Indeed, ATP5I KO cells exhibit $EC_{50}$ values up to nine times higher than those of control cells (*Figure 3H*). Overall, ATP5I KO phenocopies metformin activity and blunts the additional effect of metformin, suggesting that metformin acts partially through ATP5I.

Similar trends were obtained when using 100 μM phenformin in Seahorse kinetic with more pronounced changes in both respiratory activity decline (*Figure 3I*) and glycolysis activation (*Figure 3J*). These differences appear earlier but over a shorter time compared to metformin, possibly due to phenformin's pharmacokinetic profile (*Segal et al., 2011*). Again, phenformin affects both control and ATP5I KO cells, but it does so less efficiently in ATP5I KO cells. The difference in OCR/ECAR ratio between control cells and the two ATP5I KO cell lines at 3 hr is significantly less impacted (*Figure 3K*), indicating lesser sensitivity of ATP5I KO cells to phenformin. Resistance to phenformin is also evident in cell growth, with ATP5I KO cells showing $EC_{50}$ up to four times higher than those of control cells (*Figure 3L*). These results suggest that ATP5I mediates the metabolic and antiproliferative effects of biguanides in KP-4 cells.

## Re-expression of ATP5I normalizes mitochondrial morphology, metabolic profile, and resensitizes ATP5I knockout pancreatic cancer cells to biguanides

To validate our previous hypotheses regarding the mechanism of action of metformin, we reintroduced wild-type ATP5I in ATP5I KO KP-4 clones. Immunoblot analysis results confirmed that exogenous ATP5I could be re-expressed in ATP5I KO cells, albeit at slightly reduced levels compared to control cells. Reintroducing ATP5I also restored the protein levels of ATP5L, OSCP, NDUFB8, and COX II in KP-4 ATP5I KO cells (*Figure 4A*). Furthermore, exogenous ATP5I colocalized with TOMM20, indicating its successful mitochondrial localization and led to a reorganization of the mitochondrial networks (*Figure 4—figure supplement 1*). Indeed, quantitative analysis showed that ATP5I KO cells with exoATP5I exhibited an intermediate fragmented phenotype between punctate and filamentous forms (*Figure 4B–F*).

Similarly, ATP5I KO cells expressing exogenous ATP5I showed increased $NAD^+/NADH$ ratios (*Figure 5A*) by restoring $NAD^+$ concentration (*Figure 5—figure supplement 1A, B*), increased OCR/ECAR ratio (*Figure 5B*), enhancing respiration and reducing compensatory glycolysis (*Figure 5—figure supplement 1C, D*) to levels comparable to control cells. This metabolic reorganization made these cells up to 3 times less sensitive to 2-D-deoxyglucose (*Figure 5—figure supplement 1E*) compared to ATP5I KO cells, thereby alleviating energy stress as seen with less phosphorylated AMPK (*Figure 5C*), a tendency to increasing ATP levels (*Figure 5D*), and enabling increased cell growth (*Figure 5E*). Of

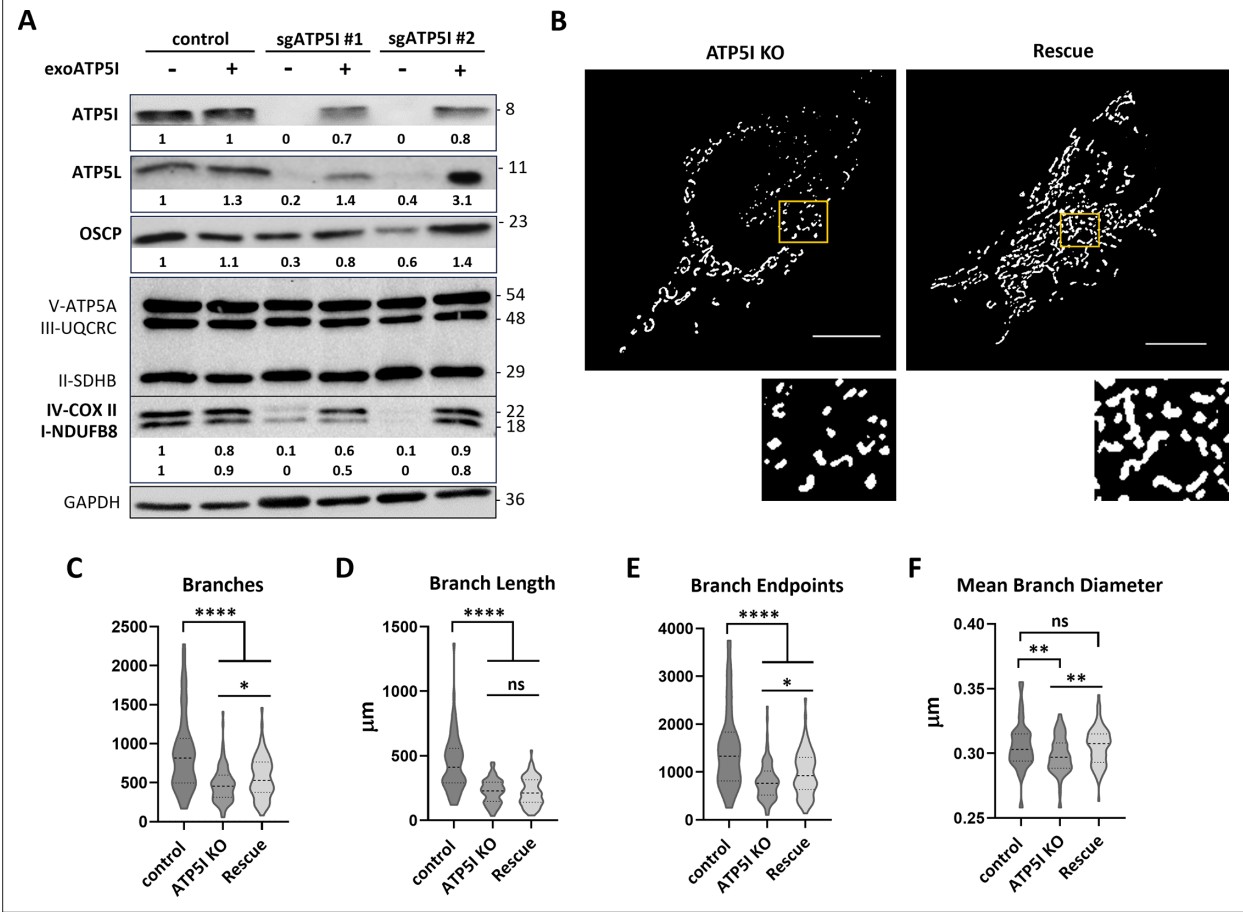

**Figure 4.** Exogenous ATP5I enables the reorganization of mitochondrial network in ATP5I knockout pancreatic cancer cells. (**A**) A representative immunoblots for the indicated proteins in KP-4 cells expressing exogenous ATP5I (exoATP5I: +) in control cells expressing a small guide RNA against GFP (sgGFP) or in ATP5I KO cells (clones of two different small guide RNAs: sgATP5I #1 sgATP5I #2) compared with the same cell lines without expression of exogenous ATP5I (-). GAPDH antibody was used as loading control. (**B**) Representative threshold images of mitochondrial morphologies visualized by TOMM20 immunofluorescence (scale bar = 10 μm) of ATP5I KO cells and their derivative re-expressing ATP5I. A magnified inset (yellow box) is shown for each image to highlight mitochondrial structural details. All images were analyzed using the Mitochondria Analyzer plugin in Fiji (ImageJ). Quantitative analysis of key mitochondrial parameters: (**C**) number of branches, (**D**) total branch length, (**E**) number of branch endpoints, and (**F**) mean branch diameter. Data represent mean ± standard deviation from $N = 3$ independent clones, with 50–100 cells analyzed per clone. ns: not significant, **$p < 0.01$, ****$p < 0.0001$ using an unpaired Student's $t$-test.

The online version of this article includes the following source data and figure supplement(s) for figure 4:

**Source data 1.** PDF file containing original western blots for *Figure 4*.

**Source data 2.** Contain original TIF files used to make *Figure 4*.

**Figure supplement 1.** Exogenous ATP5I restores mitochondrial networks in ATP5I-deficient KP-4 cells.

note, ATP levels are not significantly different between the KO and control cells perhaps because of efficient compensatory glycolysis. Restoring the ATP5I expression is also correlated with increased sensitivity to the metabolic effects induced by metformin and phenformin on mitochondrial respiration and glycolysis and decreased OCR/ECAR ratios (*Figure 5—figure supplement 2*). The rescue also enhanced the antiproliferative effects of metformin, with $EC_{50}$ values up to fivefold lower than those of ATP5I KO cells (*Figure 5F*), as well as those of phenformin, with $EC_{50}$ values up to threefold lower than those of ATP5I KO cells (*Figure 5G*).

In conclusion, these findings collectively demonstrate that the re-expression of ATP5I in ATP5I KO clones rescues mitochondrial morphology, metabolic profile, and sensitivity to biguanides, which supports the concept that ATP5I mediates the metabolic and antiproliferative effects of these compounds in KP-4 cells.

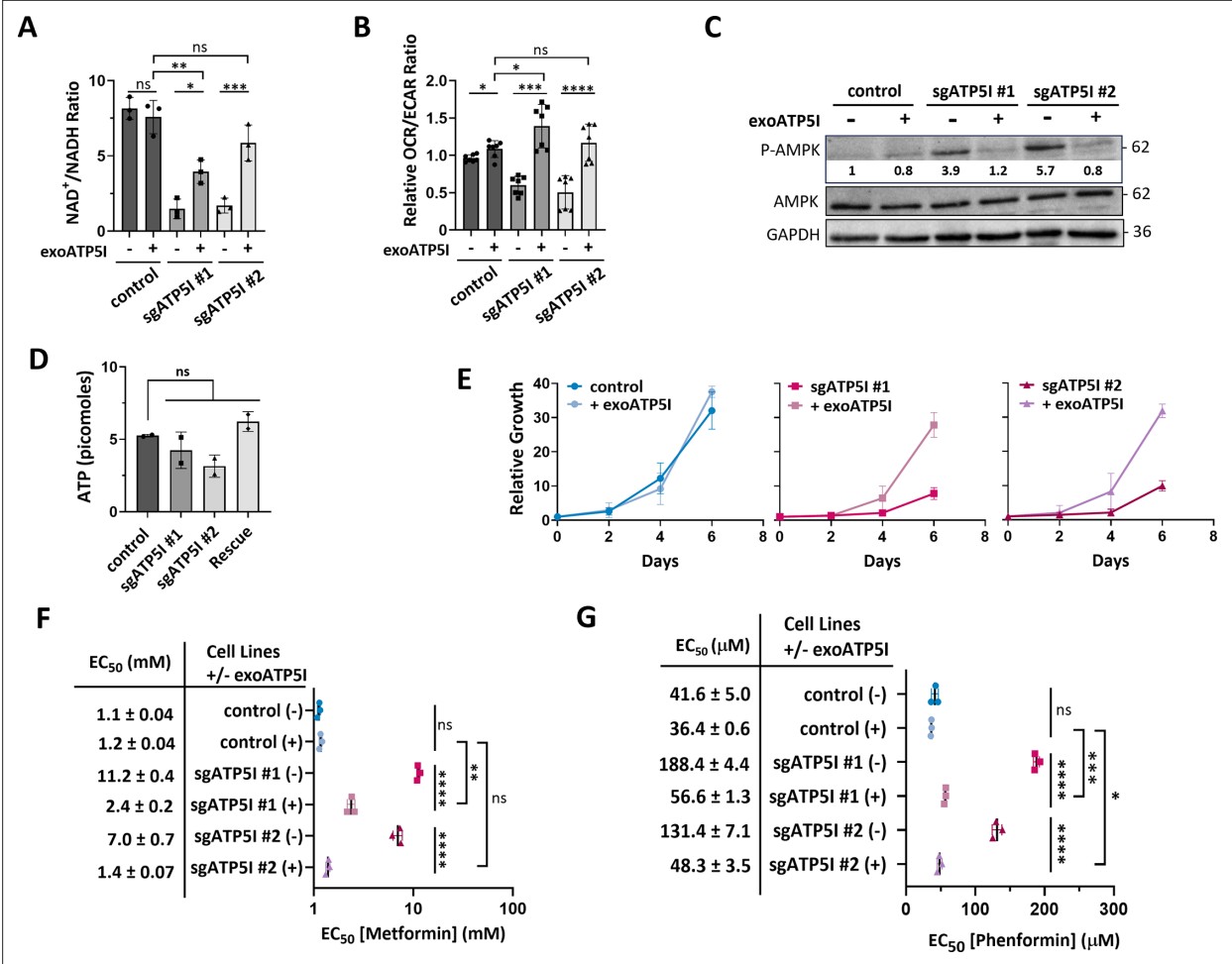

**Figure 5.** Re-expression of ATP5I rescues metabolic profile and resensitizes ATP5I knockout pancreatic cancer cells to biguanides. (**A**) Quantification of NAD$^+$/NADH ratio in KP-4 cells expressing exogenous ATP5I (exoATP5I: +) in control sgGFP or a representative clone of two different small guide RNAs (sgATP5I #1 and sgATP5I #2) compared with the same cell lines without expression of exogenous ATP5I (-). Values represent the mean ± standard deviation of three biological replicates. ns: not significant, *p < 0.05, **p < 0.01, ***p < 0.001 using an ordinary one-way ANOVA with Sidak's multiple comparison test. (**B**) Relative quantification of oxygen consumption rate (OCR) over extracellular acidification rate (ECAR) by Seahorse analysis in cells as in (**A**). Values represent the mean ± standard deviation of at least three biological replicates. ns: not significant, *p < 0.05, ***p < 0.001, ****p < 0.0001 using a repeated measures (RM) one-way ANOVA with Sidak's multiple comparison test. (**C**) Immunoblot of total and phosphorylated levels of AMPK (Thr172) protein in extracts from cells as in (**A**). GAPDH was used as loading control. (**D**) Intracellular ATP levels measured in cell lines as in (**A**). Data are presented as mean ± standard deviation. N = 2. ns: not significant. (**E**) Growth curves of cells as in (**A**) by measuring the relative number of cells over 6 days. Media was changed every 2 days. (**F**) EC$_{50}$ values of metformin (Met) treatments in cells as in (**A**). Values represent the mean ± standard deviation of N = 3. ns: not significant, **p < 0.01, ****p < 0.0001 using an ordinary one-way ANOVA with Sidak's multiple comparison test. (**G**) EC$_{50}$ values of phenformin (Phen) treatment in cells as in (**A**). Values represent the mean ± standard deviation of three biological replicates. ns: not significant, *p < 0.05, ***p < 0.001, ****p < 0.0001 using an ordinary one-way ANOVA with Sidak's multiple comparison test.

The online version of this article includes the following source data and figure supplement(s) for figure 5:

**Source data 1.** PDF file containing original western blots for *Figure 5*.

**Source data 2.** Contain original TIF files used to make *Figure 5*.

**Figure supplement 1.** Exogenous ATP5I rescues NAD metabolism, mitochondrial respiration, glycolytic compensation, and 2-deoxyglucose sensitivity in ATP5I-deficient KP-4 cells.

**Figure supplement 2.** ATP5I is required for biguanide-induced remodeling of mitochondrial respiration and glycolytic flux in KP-4 cells.

## The Seahorse bioenergetic test reveals additional similarities between biguanides and ATP5I KO

To gain a deeper understanding of the individual contributions of respiratory complexes and the $F_1F_0$-ATP synthase in the cellular response to biguanides, we conducted the Seahorse Bioenergetic

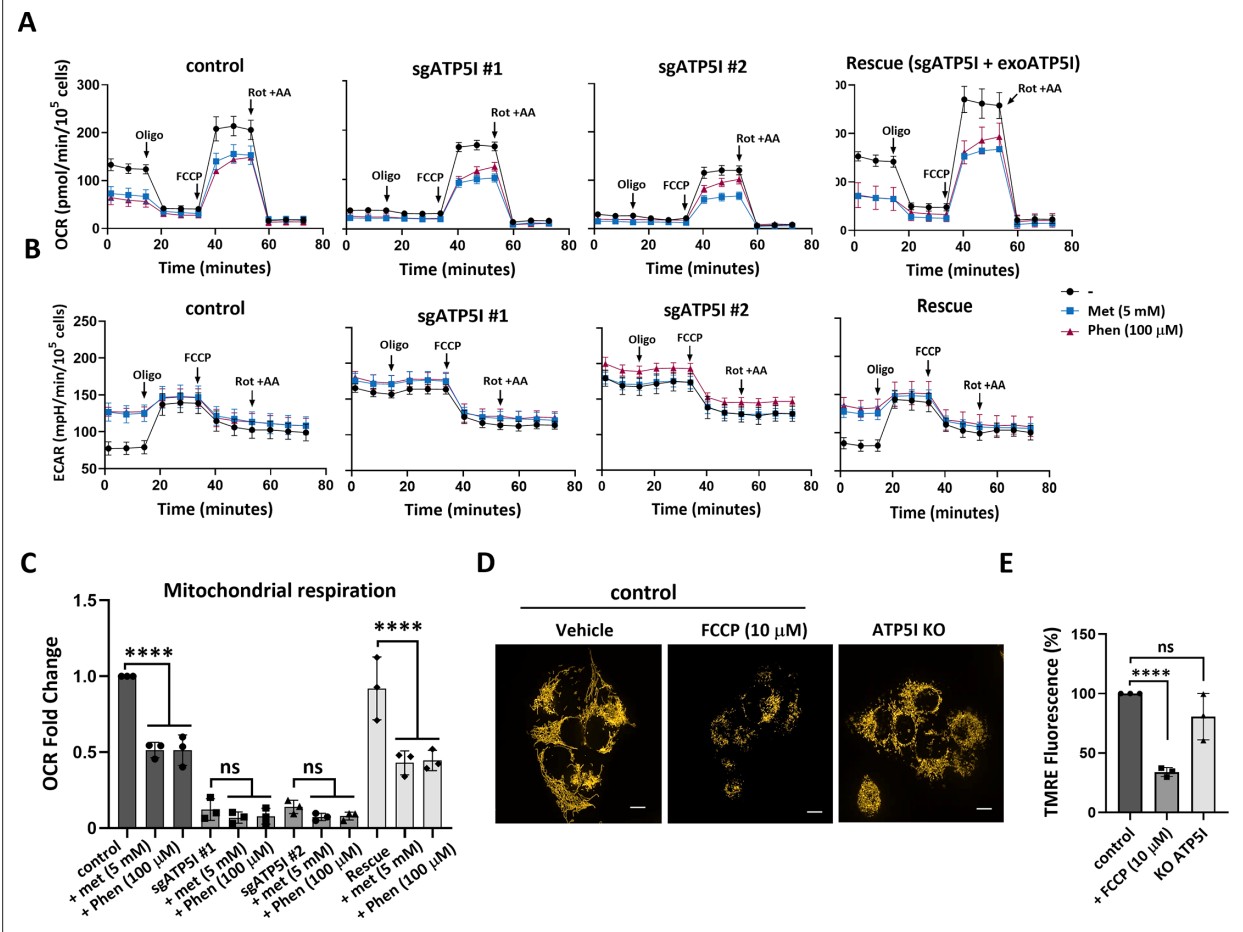

**Figure 6.** ATP5I deletion mimics biguanide-induced bioenergetic remodeling. (**A**) Representative oxygen consumption rate (OCR) profiles following sequential injection of oligomycin (Oligo), FCCP, and rotenone/antimycin A (Rot/AA) in control, ATP5I knockout (sgATP5I #1 and #2), and Rescue (sgATP5I+exoATP5I) cells treated with vehicle, 5 mM metformin (Met), or 100 µM phenformin (Phen). (**B**) Representative extracellular acidification rate (ECAR) profiles under the same conditions as in (**A**). (**C**) Quantification of basal mitochondrial respiration corresponding to the conditions in (**A**), $N = 3$. (**D**) Representative confocal images of cells stained with the membrane potential sensitive dye TMRE (100 nM, 30 min at 37°C in complete DMEM without phenol red) under control conditions, following ATP5I knockout (ATP5I KO), or after depolarization with FCCP (10 µM, 30 min prior to TMRE incubation), scale bar = 10 µm. (**E**) Quantification of TMRE fluorescence intensity per cell. Data are expressed as mean ± standard deviation and normalized to control levels. Results are from three independent experiments performed on separate days ($n = 431$ cells for control; $n = 371$ for FCCP; $n = 536$ for ATP5I KO). ns: not significant, ***p < 0.001$, ****p < 0.0001$ using an unpaired Student's $t$-test.

Mito Stress Test. This test measures baseline OCR and the response to specific inhibitors of mitochondrial complexes. Under basal conditions, treatment with biguanides at the indicated concentrations for 3 hr reduced the OCR in control cells, but less so in ATP5I KO cells, where OCR was already downregulated. Adding back ATP5I restored respiration and biguanide sensitivity to ATP5I KO cells (**Figure 6A**). Inhibition of the $F_1F_0$-ATP synthase with oligomycin reduced the OCR in control cells, but less so in biguanide-treated cells or in ATP5I KO cells (**Figure 6A**). Following treatment with the uncoupler FCCP, respiration increases in biguanide-treated cells, and to a lesser extent in ATP5I KO cells (**Figure 6A**). Adding rotenone and antimycin A totally blocks this respiration confirming that FCCP requires active complexes I–IV to induce maximal respiration. Since defects in complex I and other respiratory complexes are characterized by a lack of response to uncouplers (**Arroum et al., 2023**; **Schöckel et al., 2015**), our results are consistent with either a direct but partial inhibitory activity of biguanides on the complexes or targeting the $F_1F_0$-ATP synthase by biguanides and a subsequent disorganization of the electron transport chain and cristae (indirect mechanism). These effects are even more pronounced in ATP5I KO cells, where $F_1F_0$-ATP synthase oligomerization is completely inhibited (**Figure 2G**).

Analysis of ECAR reveals elevated basal glycolytic activity both in biguanide-treated and in ATP5I KO cells (*Figure 6B*). Moreover, while oligomycin typically increases glycolysis in control cells by inhibiting $F_1F_0$-ATP synthase and promoting compensatory glycolytic ATP production, it has little to no effect in ATP5I-deficient cells or in biguanide-treated cells (*Figure 6B*). This suggests that biguanide treatment or ATP5I loss by impairing complex V function render cells less responsive to oligomycin. Importantly, the reintroduction of ATP5I (rescue model) restored glycolytic activity to levels comparable to those of control cells (*Figure 6B*). Of note, biguanides inhibited OCR by 50% in control cells but did not significantly reduce respiration in ATP5I KO cells (*Figure 6C*). Adding back ATP5I to KO cells restored the biguanide sensitivity of OCR (*Figure 6C*).

Finally, one could argue that biguanides have a lesser effect in ATP5I KO cells due to a disruption in membrane potential which is required for metformin to enter mitochondria. We thus used TMRE (tetramethylrhodamine ethyl ester) to measure the mitochondrial membrane potential in ATP5I KO cells and found it to be indistinguishable from that of control cells expressing wild-type ATP5I, although markedly higher than in cells treated with the uncoupler FCCP that reduced it by 65% (*Figure 6D, E*). These results align with the morphological changes detailed in *Figure 2* and indicate that ATP5I deletion compromises mitochondrial structure without causing a profound loss of membrane polarization. Together, these findings indicate that while both FCCP treatment and ATP5I deletion disrupt mitochondrial morphology, only FCCP leads to a pronounced loss of membrane potential. These results support our proposition that biguanides require ATP5I to inhibit mitochondrial function.

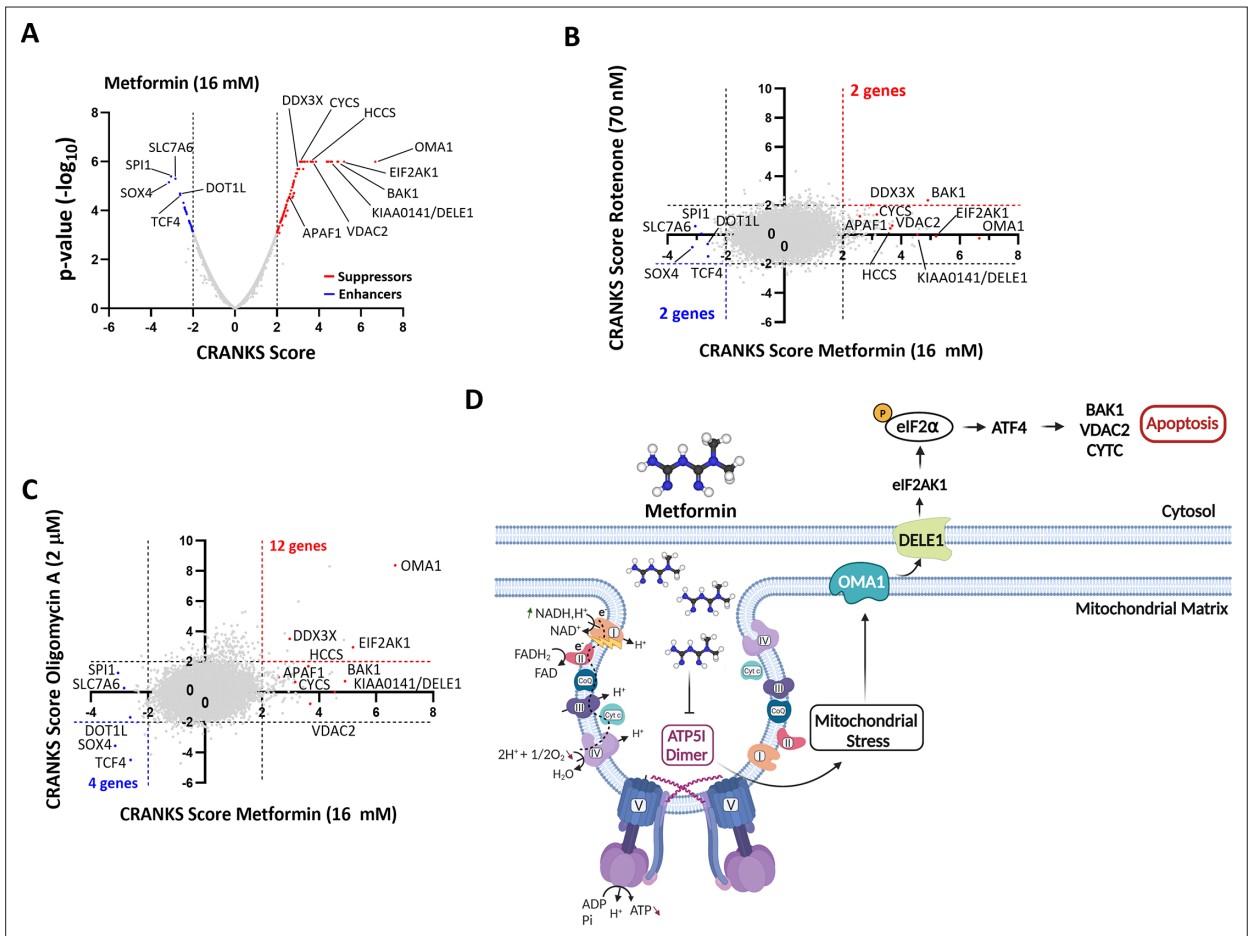

**Figure 7.** Chemogenomic screening of metformin reveals an imprint on $F_1F_0$-ATP synthase. (**A**) Results of the pooled genome-wide CRISPR/Cas9 KO screen made in NALM-6 cells treated with 16 mM metformin or control. Data are represented as a Volcano plot of gene enrichment/depletion scores vs p-values from using the CRANKS algorithm. Some genes of interest are labeled. Enhancers of metformin growth inhibition with negative CRANKS scores below 2.5 (dashed line) are labeled blue, while suppressors with positive CRANKS scores above 2.5 (dashed line) are labeled red. (**B, C**) Pairwise comparison of gene CRANKS scores obtained from screening metformin 16 mM in NALM-6 cells against that from screening either 70 nM rotenone or (**C**) 2 μM oligomycin A. (**D**) Model for metformin action triggering the OMA1–DELE1–HRI pathway.

# Chemogenomic screening of metformin reveals an imprint on $F_1F_0$-ATP synthase

Several candidate targets have been reported for biguanides, and our results presented so far suggest a new one. Clues about drug mechanism of action can be obtained in an impartial manner using genetic perturbation (*Bruno et al., 2017*). To obtain unbiased observation of biological processes affected by metformin, we performed a genome-wide pooled CRISPR/Cas9 KO screen in NALM-6 cells cultured in the presence of metformin at a concentration affecting growth (16 mM). Over the 8-day period of screening, cells had 1.47 population doublings, whereas untreated controls had 7.5 (a 65-fold difference in growth). Based on the sgRNA frequency changes observed afterward between treated vs untreated samples, we generated enrichment/depletion scores for each gene using the CRANKS algorithm. Many gene KOs were then predicted to potentiate (sgRNA depletion, KO creates sensitivity, negative CRANKS scores) or to suppress (sgRNA enrichment, KO creates resistance, positive CRANKS scores) metformin-induced growth inhibition (*Figure 7A*). Inactivation of the amino acid transporter SLC7A6 that plays a role in arginine export (*Bröer et al., 2000*) enhanced metformin activity, likely because this protein may also export metformin. Indeed, the end of the R group is very similar chemically to metformin. In addition, disabling the transcription factors SOX4, TCF4, SPI1, and the histone H3K79 methyltransferase DOT1L also enhanced metformin activity, perhaps because they promote gene expression programs that compensate for the mitochondrial dysfunction associated with metformin action. For example, SOX4 and SPI1 can increase glycolysis by promoting HK2 expression (*Khanna et al., 2023*; *Liu et al., 2023*), while TCF4 increases glycolysis by inducing the expression of the glucose transporter GLUT3 (*Liu et al., 2020*). Moreover, metformin can increase DOT1L-mediated H3K79 methylation, promoting the expression of SIRT3, a mitochondrial deacetylase that stimulates mitochondrial health and biogenesis (*Karnewar et al., 2018*). On the other hand, many genes were found to be required for metformin-mediated growth inhibition. Of note, DDX3X, an RNA helicase required for the translation of interferon response genes (*Sharma et al., 2023*), suggests a role for the interferon pathway in the antiproliferative activity of metformin. The actions of metformin were also suppressed by inactivation of the OMA1–DELE1–HRI pathway. OMA1 is a protease localized to the inner mitochondrial membrane activated by mitochondrial stress. DELE1 is a target of OMA1 that interacts with and activates EIF2AKI (also known as heme-regulated inhibitor of HRI). HRI catalyzes eIF2a phosphorylation, which in turn activates the translation of ATF4 (*Guo et al., 2020*), which is also stimulated by DDX3X (*Adjibade et al., 2017*). ATF4, together with ATF3, regulates the expression of pro-apoptotic genes in response to both ER and mitochondrial stress (*Guo et al., 2020*; *Chen et al., 2022*). Finally, inactivation of VDAC2, BAK1, HCCS (which makes Cytochrome C), Cytochrome C (CYCS), and APAF1 also conferred resistance to metformin, suggesting that metformin induces the classical mitochondria-related intrinsic apoptosis in treated cells. We compared the pattern of genetic interactions of metformin with that of rotenone, a complex I inhibitor (*Figure 7B*), and oligomycin A, a complex V inhibitor (*Figure 7C*). Interestingly, there were more similarities between oligomycin A and metformin (16 genes in common) than between rotenone and metformin (4 genes in common). Notably, inactivation of the mitochondrial apoptotic pathway through OMA1–DELE1–HRI, as well as the glycolytic pathway through SOX4 and TCF4, significantly affected both oligomycin A and metformin but not rotenone. On the other hand, DDX3X was required for the actions of all three drugs. Taken together, this genetic experiment shows that the OMA1–DEL1–HRI pathway mediates the antiproliferative activity of biguanides or the $F_1F_0$-ATP synthase inhibitor oligomycin (*Figure 7D*), while a compensatory glycolysis protects the cells.

## Discussion

Using a biologically active, BFB, we identified and characterized the ATP synthase subunit e (ATP5I; complex V) as a candidate mitochondrial target underlying the antineoplastic activity of medicinal biguanides. Biguanides bound purified ATP5I with low-micromolar affinity, and in cells, they disrupted $F_1F_0$-ATP synthase oligomerization, accompanied by the appearance of lower-molecular-weight vestigial intermediates. Consistent with complex V involvement, biguanide-treated cells were able to reinitiate respiration upon uncoupling, and this recovered respiration was abolished by complex I and III inhibitors, supporting a model in which biguanides impair ATP synthase organization/function while leaving upstream electron transport capacity intact.

Notably, several studies have argued that complex I inhibition accounts for the mitochondrial effects of biguanides. In one report, uncouplers restored respiration in metformin-treated cells—an observation we also make here—and the authors proposed that uncoupling prevents metformin entry into mitochondria by collapsing the membrane potential required for its uptake (*Wheaton et al., 2014*). However, we observed that uncoupling also restored respiration in phenformin-treated cells, even though phenformin does not require membrane potential to accumulate within mitochondria. Another study similarly found that uncouplers restored respiration in metformin-treated pancreatic cancer cells grown in adherent conditions, but not when the same cells were grown as spheroids, where the uncoupler failed to stimulate respiration, as happens in cells treated with rotenone (*Sancho et al., 2015*). These context-dependent findings are compatible with the idea that electron transport is influenced not only by the intrinsic activity of respiratory complexes but also by their organization within the inner mitochondrial membrane and cristae—an architecture that depends on $F_1F_0$-ATP synthase oligomerization.

We noticed that biguanides have an immediate effect on mitochondrial respiration, but we needed to expose cells for longer times (72 hr in KP-4 cells) to observe an accumulation of vestigial intermediates of the $F_1F_0$-ATPase. Although the accumulation of these intermediates is consistent with biguanides affecting ATP5I functions, the explanation for the inhibition of respiration is still unknown. It is plausible that biguanides perturb functions of the peripheral stalk of the $F_1F_0$-ATP synthase, a structurally essential but still poorly characterized module with potentially broad effects on mitochondrial electron transport and respiratory activity. In addition, ATP5I has been reported to interact with and modulate proteins involved in Complex I import and assembly, including TIMMDC1 (*Guarani et al., 2014*; *Fang et al., 2021*), as well as TMEM70 and TMEM242 (*Carroll et al., 2021*). It is therefore plausible that biguanides could impair Complex I indirectly by disrupting ATP5I-dependent organization and/or the function of ATP5I partners. Consistent with this model, both biguanide treatment and ATP5I knockout lead to a reduced $NAD^+/NADH$ ratio, indicating functional impairment of mitochondrial complex I.

Inhibition of respiration can disrupt the mitochondrial membrane potential, which is essential—among other functions—for the accumulation of metformin within mitochondria. Hence, a loss of membrane potential could explain the resistance to metformin in ATP5I KO cells. However, inhibition of respiration can trigger the $F_1F_0$-ATPase to operate in reverse, utilizing glycolytic ATP to sustain the membrane potential and thereby allowing continued mitochondrial uptake of metformin (*Wheaton et al., 2014*). We confirmed this interpretation since both biguanide-treated cells and ATP5I KO cells exhibit compensatory glycolysis to maintain ATP levels close to normal, and the mitochondrial membrane potential we measured in ATP5I KO cells was similar to control cells. In addition, phenformin does not require cationic transporters and the membrane potential to inhibit mitochondrial functions (*Hawley et al., 2010*), and our ATP5I KO cells were also resistant to this drug (*Figure 5G*).

Similar to medicinal biguanides' effects on KP-4 cells (*Hébert et al., 2021*), ATP5I depletion leads to mitochondrial structural changes such as mitochondrial fragmentation. Additionally, ATP5I deletion disrupts energy metabolism, decreasing complex I-mediated NADH reoxidation and respiration, promoting glycolysis and AMPK activation, and impairing cell growth. Furthermore, ATP5I KO cells exhibit resistance to metformin and phenformin growth effects, indicating its pivotal role in mediating their metabolic and antiproliferative effects. Reintroducing wild-type ATP5I in ATP5I KO cells partially restored their sensitivity to biguanides, consistent with the proposed role of ATP5I being a biguanide target. Of note, the phenotypes observed in the absence of subunit g (ATP5L) (*Carrer et al., 2021*), the direct partner of ATP5I, and subunit f (ATP5K) (*Galber et al., 2021*) in HeLa cells are similar to those of ATP5I mutants and metformin-treated cells. Further work is needed to determine how metformin affects the interactions between ATP5I, ATP5L, and ATP5K. Together, these three proteins are located at the base of $F_1F_o$-ATP synthase peripheral stalk, which interacts with OSCP (oligomycin sensitivity conferring protein) located in the $F_1$ portion of the enzyme (*Pinke et al., 2020*). ATP5I's movement within $F_1F_o$-ATP synthase (*Pinke et al., 2020*) may regulate and transfer information between the $F_1$ and the $F_o$ rotating complex (*Davies et al., 2011*; *Guarani et al., 2014*; *Fang et al., 2021*; *Carroll et al., 2021*). These structural connections could underpin the results reported here that metformin and oligomycin share a common genetic signature in a CRISPR screening in NALM-6 cells.

Our findings, combined with recent published results, strengthen the correlation between mitochondrial ultrastructure and respiratory activity (*Gerle, 2020*), suggesting that medicinal biguanides

partially affect mitochondrial respiratory chain activity by indirectly inhibiting complex I of OXPHOS through ATP5I binding, although the exact complete molecular mechanism remains unclear. Interestingly, in our screening, metformin's effects on cell growth were inhibited by the knockout of the mitochondrial protease OMA1. The expression of the complex I protein NDUFB8 was reduced in our ATP5I KO cells. Since NDUFB8 is a target of the mitochondrial protease ClpP, whose activation can inhibit pancreatic adenocarcinoma growth (*Wang et al., 2022*), these results suggest the intriguing possibility that biguanides may exert anticancer effects by activating mitochondrial proteases.

The upregulation of ATP5I observed in hypoxia (*Levy and Kelly, 1997*), DNA damage (*Fornace et al., 1988*), liver cancer (*Chen et al., 1998*), and low-fat diet responses (*Elliott et al., 1993*; *Swartz et al., 1996*) underscores a regulatory role still poorly understood. In another study, RNA-seq data from A549 cells also suggest ATP5I levels influence metformin sensitivity (*Seo et al., 2023*), highlighting its potential clinical relevance. Identifying ATP5I as a potential antineoplastic target for medicinal biguanides provides insights into the mechanisms underlying their anticancer effects and opens new avenues for targeting mitochondrial metabolism. These findings also underscore the need for further exploration of ATP5I's functions within $F_1F_0$-ATP synthase and its connections with other OXPHOS complexes, potentially leading to novel therapeutic strategies for cancer treatment.

# Materials and methods

## Key resources table

| Reagent type (species) or resource | Designation | Source or reference | Identifiers | Additional information |
|---|---|---|---|---|
| Gene (*Homo sapiens*) | ATP5ME/ATP5I | GenBank | Gene ID: 521 | Cloned by RT-PCR |
| Strain, strain background (*Escherichia coli*) | Rosetta *E. coli* BL21 | Addgene | Bacterial strain #176583 | Competent cells |
| Cell line (*Homo sapiens*) | KP-4 | Dr. N. Bardeesy | RRID:CVCL_1338 | Pancreatic ductal carcinoma |
| Cell line (*Homo sapiens*) | U2OS | ATCC | RRID:CVCL_0042 | Osteosarcoma |
| Cell line (*Homo sapiens*) | NALM-6 | Dr. M. Tyers | RRID:CVCL_UJ05 | Acute lymphoblastic leukemia (CRISPR screen) |
| Chemical compound, drug | Metformin hydrochloride | Combi-Blocks | Cat# ST-9194 | |
| Chemical compound, drug | Phenformin hydrochloride | Sigma-Aldrich | Cat# P7045 | |
| Chemical compound, drug | 2-deoxy-D-glucose | BioShop | Cat# DXG498.5 | |
| Chemical compound, drug | D-biotin | Thermo Fisher Scientific | Cat# B1595 | |
| Chemical compound, drug | Rotenone | Sigma-Aldrich | Cat# R8875 | Mitochondrial inhibitor |
| Chemical compound, drug | Oligomycin A | Tocris Bioscience | Cat# 4110 | Mitochondrial inhibitor |
| Chemical compound, drug | TMRE | Thermo Fisher Scientific | Cat# T669 | Mitochondrial membrane potential dye |
| Chemical compound, drug | MitoTracker Green | Thermo Fisher Scientific | Cat# M46750 | Mitochondrial mass control |
| Chemical compound, drug | FCCP | Abcam | Cat# NC0474145 | Mitochondrial uncoupler |
| Chemical compound, drug | Coomassie Brilliant Blue G-250 | Bio-Rad | Cat# 1610406 | Native-PAGE |
| Chemical compound, drug | Coomassie Brilliant Blue R-250 | Thermo Fisher Scientific | Cat# 33445225GM | Protein staining |
| Chemical compound, drug | Crystal violet stain | Sigma-Aldrich | Cat# B21932.14 | Cell viability assay |
| Chemical compound | Vectashield mounting medium with DAPI | Vector Laboratories | Cat# H-1200-10 | Nuclear staining |
| Antibody | anti-phospho-ACC S79 (Rabbit polyclonal) | Cell Signaling | Cat# 3661S RRID:AB_330337 | WB (1:1000) |
| Antibody | anti-AMPK (Rabbit polyclonal) | Cell Signaling | Cat# 2532 RRID:AB_330331 | WB (1:1000) |

*Continued on next page*

*Continued*

| Reagent type (species) or resource | Designation | Source or reference | Identifiers | Additional information |
|---|---|---|---|---|
| Antibody | anti-phospho-AMPK T172 (Rabbit polyclonal) | Cell Signaling | Cat# 2531 RRID:AB_330330 | WB (1:1000) |
| Antibody | anti-ATP5I (Rabbit polyclonal) | Proteintech | Cat# 16483-1-AP RRID:AB_2062052 | WB (1:500–1000), IF (1:100) |
| Antibody | anti-OXPHOS cocktail (Mouse monoclonal) | Abcam | Cat# ab110411 RRID:AB_2756818 | WB (1:750) |
| Antibody | anti-F1-ATPase β-subunit (Mouse monoclonal) | Sigma-Aldrich | Cat# MABS1304 | WB (1:1000), BN-PAGE |
| Antibody | anti-OSCP (Mouse monoclonal) | Abcam | Cat# ab110276 RRID:AB_10887942 | WB (1:1000) |
| Antibody | anti-ATP5L (Rabbit polyclonal) | Abcam | Cat# ab126181 RRID:AB_11129974 | WB (1:1000) |
| Antibody | anti-GAPDH (Goat polyclonal) | Novus Biologicals | Cat# NB300-320 RRID:AB_10001796 | WB (1:3000) |
| Antibody | anti-α-Tubulin (Mouse monoclonal) | Sigma-Aldrich | Cat# T6074 RRID:AB_477582 | WB |
| Antibody | anti-β-Actin (Mouse monoclonal) | Cell Signaling | Cat# 3700 RRID:AB_2242334 | WB (1:10000) |
| Antibody | anti-TOMM20 (Mouse monoclonal) | Santa Cruz Biotechnology | Cat# sc-17764 RRID:AB_628381 | IF (1:100) |
| Antibody | anti-rabbit IgG HRP (Goat polyclonal) | Bio-Rad | Cat# 170-6515 RRID:AB_11125142 | Secondary WB (1:3000) |
| Antibody | anti-mouse IgG HRP (Goat, clonality unknown) | Bio-Rad | Cat# 170-6516 RRID:AB_11125547 | Secondary WB (1:3000) |
| Antibody | anti-goat IgG HRP (Donkey polyclonal) | Santa Cruz Biotechnology | Cat# sc-2020 RRID:AB_631728 | Secondary WB |
| Antibody | anti-mouse AF488 (Goat polyclonal) | Invitrogen | Cat# A-11029 RRID:AB_2534088 | IF (1:2000) |
| Antibody | anti-rabbit AF568 (Goat, clonality unknown) | Invitrogen | Cat# A-11011 RRID:AB_143157 | IF (1:2000) |
| Other | Streptavidin AF488 | Invitrogen | Cat# S11223 RRID:AB_2336881 | Fluorescent detection of biotinylated proteins |
| Recombinant DNA reagent | pET-TEV-ATP5I (plasmid) | Dr. J.G. Omichinski | N/A | N-terminal 6xHis-tag expression |
| Recombinant DNA reagent | lentiCRISPRv2 (plasmid) | Addgene | Cat# 52961 | CRISPR/Cas9 vector |
| Recombinant DNA reagent | MSCV-ATP5I (plasmid) | This paper | N/A | Retroviral expression |
| Recombinant DNA reagent | pCW-Cas9 (plasmid) | Addgene | Cat# 50661 | Inducible Cas9 expression |
| Transfected construct (human) | sgATP5I #1 | This paper | N/A | CRISPR guide RNA targeting ATP5I |
| Transfected construct (human) | sgATP5I #2 | This paper | N/A | CRISPR guide RNA targeting ATP5I |
| Transfected construct (human) | sgATP5I #3 | This paper | N/A | CRISPR guide RNA targeting ATP5I |
| Transfected construct (human) | sgATP5I #4 | This paper | N/A | CRISPR guide RNA targeting ATP5I |
| Transfected construct (human) | sgGFP | This paper | N/A | Control guide RNA |
| Peptide, recombinant protein | ATP5I (6xHis-tagged) | This paper | N/A | Recombinant purified protein |
| Commercial assay or kit | Mitochondria Isolation Kit | Abcam | Cat# ab110170 | |
| Commercial assay or kit | NAD+/NADH assay kit | Sigma-Aldrich | Cat# MAK460 | |
| Commercial assay or kit | ATP Determination Kit | Thermo Fisher Scientific | Cat# A22066 | |

*Continued on next page*

*Continued*

| Reagent type (species) or resource | Designation | Source or reference | Identifiers | Additional information |
|---|---|---|---|---|
| Commercial assay or kit | BCA Protein Assay | Pierce | Cat# 23225 | Protein quantification |
| Software, algorithm | ImageJ / Fiji | NIH | RRID:SCR_003070 | Image analysis |
| Software, algorithm | GraphPad Prism | GraphPad | RRID:SCR_002798 | Statistical analysis |
| Software, algorithm | PEAKS 7 | Bioinformatics Solutions | N/A | Proteomics |
| Software, algorithm | Wave software | Agilent | RRID:SCR_024491 | Seahorse analysis |
| Other | Zeiss Axio-Observer Z1 spinning disk confocal microscope | Zeiss | N/A | 63× objective, Z-stack imaging |
| Other | Zeiss Axio Imager Z2 upright microscope | Zeiss | N/A | Immunofluorescence imaging |
| Other | SPARK 10 M plate reader | TECAN | N/A | Fluorescence detection |
| Other | Q-Exactive Plus | Thermo Fisher Scientific | N/A | Mass spectrometry |
| Other | Seahorse XFe96 Analyzer | Agilent | N/A | Metabolic flux |
| Other | P4SPR | Affinité Instruments | N/A | Surface plasmon resonance |

## Drugs

Metformin hydrochloride was purchased from Combi-Blocks (#ST-9194; San Diego, CA, USA), phenformin hydrochloride was purchased from Sigma-Aldrich (#P7045; Oakville, ON, Canada), 2-D-deoxyglucose was purchased from Bioshop (#DXG498.5; Burlington, ON, Canada) and D-biotin was purchased from Invitrogen/Thermo Fisher Scientific (#B1595; Waltham, MA, USA).

## Synthesis

All chemicals were purchased from Sigma-Aldrich, Oakwood Chemicals (Estill, SC, USA) and Combi-Blocks in their highest purity and were used without further purification. Deuterated dimethylsulf-oxide (DMSO-$d_6$), deuterated methanol (MeOD) and deuterium oxide ($D_2O$) were purchased from CDN Isotopes (Pointe-Claire, QC, Canada). NMR spectra were recorded on Bruker avance 400 and Bruker avance 500 spectrometers. Coupling constants ($J$) were reported in hertz (Hz), chemical shifts were reported in parts per million (ppm, $\delta$) and multiplicities were reported as singlet (s), doublet (d), triplet (t) and multiplet (m). $^1$H and $^{13}$C chemical shifts are relative to the solvents: $\delta$H 2.50 $\delta$C 39.5 for DMSO-$d_6$, $\delta$H 3.35, 4.78 and $\delta$C (49.3) for MeOD and $\delta$H 4.65 for $D_2O$. Final compounds were purified using a preparative liquid chromatography (prepLC-MS-Quadrupole, Waters) equipped with a C18 reverse phase column (2.1 × 100 mm, 3 mm, Atlantis) with HPLC-grade solvents (Sigma-Aldrich). High-resolution mass spectra and LC–MS purity analysis were collected using LC-TOF with ESI ioniza-tion source (Agilent Technologies, Santa Clara, CA, USA) by the regional mass spectrometry center of University of Montreal. After prep HPLC, all final compounds were subsequently freeze-dried and aliquots of the monochloride salt powder were carefully weighted. The detailed synthetic procedures and compound characterization are provided in the Supporting Information.

## Cell culture

Human pancreatic ductal carcinoma cell line KP-4 was as a kind gift from Dr. N. Bardeesy (Massa-chusetts General Hospital, Boston). Human embryonic kidney HEK-293T cells were obtained from Thermo Fisher Scientific. Phoenix ampho packaging cells was as a kind gift from Dr. S.W. Lowe (MSK, New York). NALM-6 cells were a gift from Stephen Elledge (Harvard, USA). U2OS cells were purchased from ATCC. Except for NALM-6 cells which were cultured in RPMI medium (#350-035; Wisent), all cells were cultured in Dulbecco's modified Eagle medium [DMEM] #319-015; (Wisent, St-Jean-Batiste, QC, Canada) supplemented with 10% fetal bovine serum (FBS, Wisent), 2 mmol/l L-glutamine (Wisent) and 1% penicillin/streptomycin sulfate (Wisent) in a humidified incubator at 37°C with 5% $CO_2$. All cell lines tested negative for mycoplasma before use in experiments. The cell lines were also authenticated before the experiment using STR profiling.

## Immunoblotting

For immunoblotting, protein extracts were obtained from 6-well cell culture dishes at 80–90% confluence. Cells were treated with the corresponding drugs or vehicle 16 hr after seeding. As previously described (*Igelmann et al., 2021*), cells were washed twice with ice-cold phosphate-buffered saline (PBS) solution, then the residual PBS was removed by aspiration. Cells were lysed in 250 µl of 2x Laemmli buffer (4% SDS, 20% glycerol, 120 mM Tris-HCl pH 6.8), recovered using a cell scraper, and transferred into Eppendorf tubes for a mild sonication of 20 s followed by heating for 5 min at 95°C. Protein extracts were cooled to room temperature (rt) and quantified by measuring absorbance at 280 nm using a Nanodrop microvolume spectrophotometer (2000c, Thermo Fisher Scientific). Samples were then diluted to a final concentration (2 mg/ml) using a modified Laemmli Buffer (10% β-mercaptoethanol, 0.1% bromophenol, 2x Laemmli) and were kept at –20°C until use. A quantity of 25–40 µg of protein extracts was loaded into SDS–PAGE (12–15% for resolving gels, 4% for stacking gels) and run in Tris-Glycine SDS buffer (192 mM glycine, 0.1% SDS, 25 mM Tris-Base) at 90 V. SDS–PAGE were then transferred on a nitrocellulose membrane (0.45 µm, Bio-Rad, Mississauga, ON, Canada) in Tris-Glycine buffer (96 mM glycine, 10 mM Tris-Base) for 1h30 at 120 V. Membranes were blocked in a modified Tris-buffered saline + 0.05% Tween-20 (TBST) solution with 5% skim milk for 1 hr at rt, washed with TBST 3 × 5 min at rt, and were then incubated with an appropriately diluted solution of primary antibody (in 0.1% bovine serum albumin [BSA], 0.02% sodium azide, PBS pH 7.4) overnight (o/n) at 4°C. After primary antibody incubation, membranes were washed with TBST 3 × 5 min and incubated with a diluted (1:3000 in 1% skim milk/TBST) secondary antibody coupled to HRP for 1 hr at rt. After secondary antibody incubation, membranes were washed with TBST 3 × 5 min. The signal was revealed using enhanced chemiluminescence reagent (#RPN2106, GE Healthcare Life Sciences, Chicago, IL, USA) and images were acquired by exposition with autoradiographic films or using ChemiDoc Imaging Systems (XRS+; Bio-Rad). FroggaBio BLUelf Prestained Protein ladder (FroggaBio, Concord, ON, Canada) was used to estimate protein molecular weight. Of note, most membranes were cut into pieces and were incubated with several antibodies. The following primary antibodies were used: anti-phospho-ACC S79 (Rabbit, 1:1000, #3661S, Cell signaling, Danvers, MA, USA), anti-AMPK (Rabbit, 1:1000, #2532, Cell signaling), anti-phospho-AMPK T172 (Rabbit, 1:1000, #2531, Cell signaling), anti-ATP5I (Rabbit, 1:500, #16483-1-AP, Proteintech, Rosemont, IL, USA), anti-OXPHOS cocktail human (Mouse, 1:750, #ab110411, Abcam, Toronto, ON, Canada), anti-F1-ATPase β-subunit (Mouse, 1:1000, #MABS1304, Sigma-Aldrich), anti-OSCP (Mouse, 1:1000, #ab110276, Abcam), anti-ATP5L (g) (Rabbit, 1:1000, #ab126181, Abcam), anti-GAPDH (Goat, 1:3000, #NB300-320, Novus Biologicals, Centennial, CO, USA), anti-α-Tubulin (Mouse, #T6074, Sigma-Aldrich) and anti-β-Actin (Mouse, 1:10,000, #3700, Cell signaling). The following secondary antibodies were used: anti-rabbit IgG conjugated to HRP (Goat, 1:3000, #170-6515, Bio-Rad), anti-mouse IgG conjugated to HRP (Goat, 1:3000, #170-6516, Bio-Rad) and anti-goat IgG conjugated to HRP (donkey, #Sc-2020, Santa Cruz Biotechnology, Dallas, TX, USA).

## Immunofluorescence

For immunofluorescence experiments, 50,000–100,000 cells were seeded on 1.5 mm thickness coverslips and were incubated for 48 hr. As previously described (*Lessard et al., 2018*), cells were washed twice with ice-cold PBS, then were fixed with a 4% paraformaldehyde solution for 10 min at 4°C and washed 3 × 10 min with a solution of PBS + 0.1 M glycine at rt with agitation. At this moment, coverslips were sometimes stored in PBS + 0.2% sodium azide at 4°C until use. If stored, 3 × 10 min washing steps with PBS were performed to remove the azide. Cells were then permeabilized with PBS + 0.2% Triton X-100 and 3% BSA for 5 min at rt. Then, cells were washed in a blocking solution (3% BSA in PBS pH 7.4) 3 × 10 min at rt with agitation and were then incubated with an appropriately diluted solution of primary antibody (in 3% BSA, PBS pH 7.4) o/n at 4°C in a humidified chamber. After primary antibody incubation, cells were washed in blocking solution 3 × 10 min and were incubated with an appropriately diluted solution (3% BSA, PBS pH 7.4) of secondary AlexaFluor antibody for 1 hr at rt in the dark. Cells were then washed with PBS 3 ×10 min and coverslips were mounted on glass coverslips in Vectashield mounting media containing DAPI (H-1200–10, Vector Laboratories, Newark, CA, USA) and were sealed off with nail polish and kept for a minimum of 24 hr at 4°C. Images were collected with the Zeiss Axio Imager Z2 upright microscope equipped with a CoolSNAP FX camera (Photometrics), Axiocam camera and ZEN 2 blue edition software and were analyzed using ImageJ software.

Colocalization was assessed using the Job Plot profile functionality of ImageJ by determining fluorescence intensity for each pixel of each channel across a line drawn. For quantifications and colocalizations, raw data were exported to Prism (10.2.2, GraphPad) software to generate final figures. The following primary antibodies were used: Anti-ATP5I (Rabbit, 1:100, #16483-1-AP, Proteintech) and TOMM20 (Mouse, 1:100, #sc-17764, Santa Cruz Biotechnology). The following secondary antibodies were used: anti-mouse AF488 (Goat, 1:2000, #A-11029, Invitrogen), anti-rabbit AF568 (Goat, 1:2000, #A-11011, Invitrogen), and Streptavidin AF488 (#S11223, Invitrogen). For automated quantifications, all images were first deconvoluted to enhance signal clarity. Mitochondrial morphology and network were then analyzed using the Mitochondria Analyzer plugin for Fiji (*Gu et al., 2022*).

## TMRE-based mitochondrial membrane potential assay

Mitochondrial membrane potential was assessed by live-cell fluorescence imaging using the cationic, lipophilic dye TMRE (#T669, Thermo Fisher Scientific). Cells were incubated with 100 nM TMRE in complete DMEM (with serum, without phenol red) for 30 min at 37°C in a humidified $CO_2$ incubator. For depolarization controls, cells were pretreated with 10 µM FCCP for 30 min prior to TMRE addition. Immediately before imaging, cells were washed twice with DMEM without phenol red (#319-080 CL, Wisent) to remove excess dye. Z-stack images were acquired from live cells using a Zeiss Axio-Observer Z1 spinning disk confocal microscope (63× objective, 12 optical slices). Three-dimensional reconstruction and quantitative analysis were performed using Fiji (ImageJ). Mitochondria were segmented using a custom macro based on Otsu's thresholding method, followed by region of interest (ROI)-based quantification of area, mean intensity, perimeter, and integrated density via the ROI Manager. To verify that changes in TMRE signal reflected alterations in membrane potential rather than mitochondrial content, cells were co-stained with 50 nM MitoTracker Green (#M46750, Thermo Fisher Scientific) under the same conditions. Since MitoTracker Green accumulates in mitochondria independently of membrane potential, it served as a control for mitochondrial mass. Results from this control experiment are not shown.

## Mitochondrial protein isolation

Mitochondrial crude extracts were obtained using Mitochondria Isolation Kit for Cultured cells (ab110170, Abcam) according to the manufacturer's instructions. For each condition, 1 × 15 cm cell culture dish of KP-4 cells was washed twice in ice-cold PBS, then the residual PBS was removed by aspiration. Cells were then scraped in ice-cold PBS and transferred in Eppendorf tubes. The following steps for mitochondrial isolation were performed as described in the manufacturer's instructions. The pellet containing mitochondrial proteins was resuspended in lauryl maltoside buffer [1% lauryl maltoside, cOmplete-EDTA free Protease Inhibitor Cocktail (Roche, Laval, QC, Canada), PhosSTOP (Roche), PBS pH 7.8]. The concentrations of mitochondrial proteins were determined using the bicinchoninic acid assay (BCA) (23225, Pierce). Of note, at the end of isolation, we obtained ~0.7–1 mg of mitochondrial proteins per 15 cm dish.

## Mitochondrial protein for BN-PAGE

Mitochondria were isolated using a procedure based on the manufacturer's protocol (Mitochondria Isolation Kit for Cultured Cells, #ab110170, Abcam), with optimized reagent volumes adapted for smaller-scale preparations. While the standard protocol recommends four 15 cm culture dishes per isolation to maximize yield, we obtained satisfactory results using only two 15 cm dishes, consistently recovering 500–1000 µg of mitochondrial protein. Briefly, cells were harvested by scraping in ice-cold 1X PBS and transferred into two pre-chilled 1.5 ml microcentrifuge tubes to balance subsequent centrifugations. Cells were pelleted by centrifugation at 1000 × *g* for 3 min at 4°C. To improve membrane disruption, a freeze–thaw cycle was performed by incubating cell pellets for 10 min on dry ice–ethanol, followed by 10 min at 37°C. Pellets were then resuspended in 500 µl of Reagent A and subjected to mechanical homogenization using a pre-chilled 2 ml glass Dounce homogenizer. Thirty strokes were applied to ensure sufficient cell disruption. The homogenate was centrifuged at 1000 × *g* for 10 min at 4°C to remove nuclei and cell debris. The supernatant was transferred to a new tube, mixed with 500 µl of Reagent B, and the protocol was followed as per the manufacturer's instructions for subsequent steps. The final mitochondrial pellet was resuspended in 250 µl of the final buffer. Two aliquots of 100 µl were reserved for downstream purification experiments, while a 50-µl

aliquot was kept for BCA quantification and OXPHOS western blot quality control. All samples were snap-frozen and stored at −80°C until use.

## Blue Native-PAGE

As part of the adapted protocol (*Brunner et al., 2002*), mitochondria (100 µg of protein) were lysed on ice for 30 min in a digitonin buffer containing 1% (wt/vol) digitonin (#11024-24-1, Sigma-Aldrich), 50 mM potassium acetate, 30 mM HEPES-KOH, 10% (vol/vol) glycerol, and EDTA-free complete protease inhibitors, pH adjusted to 7.4. Lysates were clarified by centrifugation at maximum speed for 60 min at 4°C. Coomassie Brilliant Blue G-250 (#1610406, Bio-Rad) was then added to a final concentration of 0.1% (wt/vol) prior to electrophoresis. Proteins were resolved on NativePAGE 3–12% Bis-Tris gels (#BN1001BOX, Thermo Fisher Scientific) using a discontinuous buffer system with an anode chamber (outer) and cathode chamber (inner). Electrophoresis was conducted at 4°C for 16 hr. The 20X running buffer used to prepare both the anode and cathode buffers contained 1 M Bis-Tris and 1 M Tricine, pH 6.8. The anode buffer was prepared by dilution to 1X. The 20X cathode additive contained 0.4% (wt/vol) G-250. Two working cathode buffers were used sequentially: a dark buffer with 0.02% G-250 and a light buffer with 0.002% G-250, both diluted from the 20X stock. Electrophoresis was initiated with the dark cathode buffer and anode buffer at 100 V for 2 hr at 4°C. The dark buffer was then replaced with the light buffer, and electrophoresis was continued for approximately 1 6 hr at 130 V at at 4°C. Proteins were transferred onto a PVDF membrane pre-activated in methanol for 30 s, rinsed with water, and equilibrated for 5 min in a transfer buffer containing 25 mM Bicine, 25 mM Bis-Tris, and 1 mM EDTA, pH 7.2. The transfer was performed at 100 V for 1 hr and 30 min on ice. Following transfer, the membrane was incubated in 8% (vol/vol) acetic acid for 15 min, rinsed with water, briefly treated again with methanol for 30 s, rinsed once more with water, and then blocked according to the immunoblotting procedure described in this study. Immunodetection was carried out using an anti-ATP synthase subunit β antibody (#MABS1304, Sigma-Aldrich). Signal quantification was performed with ImageJ software by measuring the area under the curve.

## Pull-down assay

For pull-down assay, 1 mg of mitochondrial proteins were incubated with either 1 mM of BFB or biotin functionalized amine (BFA) in Eppendorf tubes for 3 hr at rt with mild agitation. In parallel, Dynabeads MyOne streptavidin C1 (65001, Thermo Fisher Scientific) were washed and resuspended at least three times in lauryl maltoside (LM) buffer [1% lauryl maltoside, cOmplete-EDTA free Protease Inhibitor Cocktail (Roche), PhosSTOP (Roche), PBS pH 7.8]. After incubation with biotinylated molecules, beads (0.5 mg to 50 µl) were added to mitochondrial lysates and incubated for 30 min at 4°C with agitation. Beads were respectively recovered with a magnet, washed in LM buffer 3 × 30 min at 4°C. Elution of bound proteins from beads in BFB condition was performed by adding 30 µl of 50 mM metformin in LM buffer. After an incubation time of 30 min at rt, beads were vortexed, pelleted with a magnet and the supernatant was collected in a new tube. Proteins were then denatured by adding 30 µl of 6x SDS-loading buffer (30% glycerol, 10% SDS, 1% bromophenol blue, 15% β-mercaptoethanol, 0.5 M Tris-HCl pH 6.8) to the mixture and boiled 3 × 5 min at 95°C. Elution of bound proteins for BFA condition and remaining proteins on beads for BFB condition were achieved by boiling and vortexing the corresponding beads in 60 µl of 6x SDS loading buffer 3 × 5 min at 95°C. After denaturation of proteins, all samples were loaded into SDS–PAGE. The resulting gel was stained with Coomassie brilliant blue R-250 (#33445225GM, Thermo Fisher Scientific) then washed with a destaining solution (40% water, 50% methanol, and 10% acetic acid). Gel segments were then excised, digested with trypsin and dried overnight in a cold trap (Labconco, Kansas City, MO, USA). Three samples for each elution condition were prepared in two technical replicates and were then analyzed by the mass spectrometry platform at IRIC (Institute for Research on Immunology and Cancer). It is worth mentioning that for pull-down validation, same experiments were performed using western-blot analysis with anti-ATP5I antibody after protein denaturation steps (see Immunoblotting section).

## Proteomics

Samples were reconstituted in 50 mM ammonium bicarbonate urea 8 M vortexed and further diluted to 50 mM ammonium bicarbonate urea 1 M with 10 mM TCEP [Tris(2-carboxyethyl)phosphine hydrochloride; Thermo Fisher Scientific], and vortexed for 1 hr at 37°C. Chloroacetamide (Sigma-Aldrich)

was added for alkylation to a final concentration of 55 mM. Samples were vortexed for another hour at 37°C. Trypsin 1 µg was added, and digestion was performed for 8 hr at 37°C. Samples were dried down and solubilized in 5% ACN-4% formic acid (FA). The samples were loaded on a 1.5-µl pre-column (Optimize Technologies, Oregon City, OR). Peptides were separated on a home-made reversed-phase column (150 µm i.d. by 200 mm) with a 56 min gradient from 10 to 30% ACN-0.2% FA and a 600 nl/min flow rate on an Ultimate 3000 connected to a Q-Exactive Plus (Thermo Fisher Scientific, San Jose, CA). Each full MS spectrum acquired at a resolution of 60,000 was followed by tandem-MS (MS–MS) spectra acquisition on the most abundant multiply charged precursor ions for 3 s. Tandem-MS experiments were performed using higher energy collision dissociation at a collision energy of 30%. The data were processed using PEAKS 7 (Bioinformatics Solutions, Waterloo, ON) and a Uniprot database. Mass tolerances on precursor and fragment ions were 10 ppm and 0.01 Da, respectively. Fixed modification was carbamidomethyl (C). Variable selected posttranslational modifications were acetylation (N-ter), oxidation (M), deamidation (NQ), phosphorylation (STY).

## Constructs

For recombinant protein expression in bacteria, full-length of human ATP5I sequence was PCR-amplified and cloned using BamH1/EcoRI restriction sites into a pET-TEV vector (Addgene, Watertown, MA, USA) for the expression of an N-terminal 6x-His-tagged protein. This plasmid was obtained as a kind gift from Dr. J.G. Omichinski (University of Montreal, Montreal). For lentiviral-mediated transduction in KP-4 cells, annealed and phosphorylated oligos for two RNA guides (sgRNAs) targeting two regions of ATP5I's cDNA (sgATP51 #1 and sgATP5I #2) and one RNA guide targeting GFP (sgGFP) sequence were subcloned into the BsmBI restriction site of the lentiCRISPRv2 plasmid (#52961, Addgene). For re-expression of ATP5I in control (sgGFP) and ATP5I knockout (KO) (sgATP5I#1 and #2) cells, full-length human ATP5I sequence with modified codons (same amino acid but different nucleic acid sequence) was generated by PCR site-directed mutagenesis and cloned using BamH1/EcoRI restriction sites into the MSCV retroviral vector. All constructs were confirmed by DNA sequencing.

Constructs primers used:

| Target | 5' forward primer | 3' reverse primer |
|---|---|---|
| For recombinant protein expression: | | |
| ATP5I (ATP5ME) | GAATGGATCCGCCGCCATGGTGCCACCGGTGCA | ATCGGAATTCTCACTTTAATATGCTGTCATCTTCTGCC |
| For lentiviral mediated transduction | | |
| sgGFP | | GGGCGAGGAGCTGTTCACCG |
| sgATP5I #1 | | CGTAGGCCACACCGAGGAAC |
| sgATP5I #2 | | TGGCTCCGTAGGCCACACCG |
| For PCR site-directed mutagenesis | | |
| | GAATGGATCCGCCGCCATGGTGCCACCGGTGCA | CGCACCATATGCGACCCCTAGAAAGAGCGCGGAGTAGCGGCCGAGCTTGAT |
| | GCGCTCTTTCTAGGGGTCGCATATGGTGCGACGCGCTACAATTACCTAA | ATCGGAATTCTCACTTTAATATGCTGTCATCTTCTGCC |

## Protein expression and purification

N-terminal 6x-His-tagged ATP5I was expressed in Rosetta *E. coli* BL21 competent cells #176583, Addgene for the expression of eukaryotic proteins that contain rare codons. To overexpress recombinant protein as suggested (*Chhetri et al., 2015*), cells were grown at 37°C in terrific broth (TB) medium supplemented with 1% of ethanol (vol/vol), 100 mg/ml ampicillin, and 50 mg/ml chloramphenicol to an $OD_{600 nm}$ of 0.8. Expression was induced for 4 hr at 30°C with 1 mM isopropyl β-D-1-thiogalactopyranoside; then cells were harvested by centrifugation (15 min, 5000 rpm at 4°C). Bacterial pellets were then solubilized in buffer A (6 M guanidine, 500 mM NaCl, 20 mM $NaH_2PO_4$ pH 8) for 30 min at 4°C and lysed by sonication. Cell lysate was then harvested by centrifugation (30 min,

13,000 rpm at 4°C) and after filtration (0.45 µm) the supernatant was loaded in an immobilized metal ion affinity chromatography column (HisTrap FF, Cytiva) with buffer A. After removing non-specifically bound molecules (flow through), the column was washed with buffer B (8 M urea, 500 mM NaCl, 20 mM NaH$_2$PO$_4$ pH 8), then with buffer C (20 mM imidazole, 500 mM NaCl, 20 mM NaH$_2$PO$_4$ pH 8) until UV absorbance returned to baseline. After washing, the column was eluted with buffer D (500 mM imidazole, 500 mM NaCl, 20 mM NaH$_2$PO$_4$ pH 8) and the resulting eluate was loaded in a desalting column (HiPrep 26/10, Cytiva) with a PBS containing 125 mM NaCl, 5 mM DTT, PBS pH 7.4 and concentrated in a 3 kDa centrifugal filter (Amicon) to 1.5 mg/ml. Of note, all purification steps were done with AKTA pure chromatography system.

## Surface plasmon resonance

Experiments were performed at 25°C with a portable SPR (P4SPR) device from JF. Masson's laboratory (University of Montreal) (WO/2010/130045, Affinité instruments) using a compatible prism coated with a 200-nm streptavidin (SA) derivatized carboxymethyldextran hydrogel (Xantec bioanalytics) for immobilization of biotinylated ligands. For each solution, a volume of 300 µl was injected in the flow cells comprising a multifluidic channel cell and a reference cell. After having stabilized the baseline (~30 min) with running buffer (125 mM NaCl, 5 mM DTT, PBS pH 7.4), biotin functionalized biguanide chloride salt was immobilized on SA sensor chip at 125 µM (concentration experimentally determined to saturate streptavidin sites). Then, the chip was washed with running buffer until stabilizing the baseline. After that, several concentrations of purified recombinant ATP5I were injected until saturation of the signal. Of note, between each concentration added, washing steps were performed with running buffer to remove non-specifically bound molecules. Finally, the surface was regenerated with urea (2 M) prepared in milli-Q-water. It is worth noting that due to the low dissociation constant of biotin/streptavidin interaction, biotinylated ligands are almost irreversibly bound that make it difficult to regenerate the chip. The sensorgrams were analyzed with P4SPR Control v2.022 software and final figures were exported to Prism (10.2.2, GraphPad) using a pseudo-first order association kinetics equation for each concentration. For binding affinity curve, each steady state for each concentration were exported to Prism (10.2.2, GraphPad) software to generate the figure using a one site-specific binding equation. Of note, compounds were dissolved in the running buffer.

## Generation of stable ATP5I knockout cells

ATP5I knock out in KP-4 cell line was done with lentiviral CRISPR/Cas9 technology and sgRNAs against ATP5I or a control sgRNA (see construct section). First, $5 \times 10^6$ 293T cells were seeded in 10 cm plates and grown for 16 hr. They were then transfected using the calcium-phosphate precipitation method as previously described (*Ferbeyre et al., 2000*) with 3 µg of lentiCRISPRv2 plasmid vector, 2 µg of the *pCMV-dR8.2* plasmid, and 1 µg of the *VSV-G* envelope protein expression plasmid (both kind gift of Dr. N. Bardeesy). After 16 hr, 10 mM sodium butyrate (Sigma-Aldrich) was added for a minimum of 6 hr, and then the medium was changed. In parallel, target cells were seeded to obtain 60% confluence the following day. Media from the transfected cells were collected the following day, filtered through a 0.45-µm filter. This viral soup was supplemented with 4 µg/ml polybrene (Sigma-Aldrich) and 10% fresh media before being placed on KP-4 receiving cells. Viral soup was removed 24 hr later for fresh DMEM cell culture media (see cell culture section) supplemented with 100 µg ml$^{-1}$ sodium pyruvate (#600-110-EL, Wisent) and 50 µg ml$^{-1}$ uridine (#URD222.10, Bioshop) to help them grow in case mitochondrial functions were affected (*Adant et al., 2022*). KP-4 infected cells were selected 6 hr after media change with 2 µg/ml puromycin (#400-160-EM, Wisent). After 2 weeks in culture to ensure time alterations in the targeted regions with CRISPR/Cas9, cells were replated highly diluted to isolate clones of cells. Many clones were grown and characterized by ATP5I immunoblots to select proper ATP5I knock-out.

## Retroviral infections

ATP5I re-expression was performed by retroviral-mediated expression of wild-type ATP5I in control (sgGFP) and ATP5I KO cells (sgATP5I #1 and #2) with a MSCV-ATP5I vector (see constructs).

For retroviral infections, $5 \times 10^6$ Phoenix-Ampho packaging cells were seeded in 10 cm cell culture dishes and grown for 24 hr. Then, cells were transfected with 20 µg of MSCV-ATP5I vector and 10 µg of Helper envelope protein expression plasmid using the calcium phosphate method as previously

described (*Ferbeyre et al., 2000*). The next day, sodium butyrate was added at 10 mM for 6 hr and then fresh medium was added. In parallel, target cells were seeded to obtain 60–70% confluence on the day of infection. Two days following transfection, the viral soup from one 10 cm dish was filtered through a 0.45-μm filter, supplemented with 4 μg ml$^{-1}$ polybrene (#H-9268, Sigma-Aldrich) and 1/10 (vol/vol) of fresh media were then used to infect one 10 cm dish of target cells. After at least 12 hr post-infection, infected cells were seeded into a new dish. After at least 4 hr of adherence, infected cells were selected using 50 μg/ml hygromycin (#400-141-UG, Wisent). Of note, the selection was maintained for the length of the experiments.

## Quantitative PCR

RNA isolation and qPCR quantifications were performed as previously described (*Igelmann et al., 2021*) using the following primers (see below, BioCorp, Pierrefonds, QC, Canada). Mitochondrial to nuclear DNA ratios were performed similarly by qPCR on 30 ng of DNA purified from cultures of KP-4 cells. For the isolation, one 6 cm plate of sub-confluent KP-4 cells was rinsed twice in PBS, collected by scraping in 500 μl of SNET lysis buffer (5 mM EDTA, 400 mM NaCl, 1% SDS, 400 g/ml Proteinase K, 20 mM Tris-HCl pH 8.0 [Sigma-Aldrich]) and incubated at 55°C overnight. An equal volume of phenol:chloroform (Sigma-Aldrich) was added and left to agitate for 30 min at rt. After a 5-min centrifugation (16,000 × *g*), the aqueous phase was transferred to a new tube and precipitated by an equal volume of isopropanol, mixed well, and DNA was collected by centrifugation at maximum speed (16,000 × *g*) for 15 min at 4°C. The pellet was washed with 70% ethanol, air-dried, and resuspended in TE. qPCR primers used:

| Target | 5′ forward primer | 3′ reverse primer |
| --- | --- | --- |
| **For mRNA expression** | | |
| HMBS | GGCAATGCGGCTGCAA | GGGTACCCACGCGAATCAC |
| TBP | GCTGGCCCATAGTGATCTTTGC | CTTCACACGCCAAGAAACAGTGA |
| NDUFB8 | CCGCCAAGAAGTATAATATGCGT | GTATCCACACGGTTCCTGTTGT |
| SDHB | CACAGCTCCCCGTATCAAGAAA | TGCATGATCTTCGGAAGGTCAAA |
| UQCRC2 | TTCAGCAATTTAGGAACCACCCA | GGTCACACTTAATTTGCCACCAA |
| COXII (MT-CO2) | CCGCCATCATCCTAGTCCTCAT | GGTGGCCAATTGATTTGATGGT |
| ATP5A(ATP5F1A) | TATGACGACTTATCCAAACAGGC | CGGGAGTGTAGGTAGAACACAT |
| **For mitochondrial/nuclear ratio** | | |
| MT-DNA | GATTTGGGTACCACCCAAGTATTG | GTACAATATTCATGGTGGCTGGCA |
| Nuclear-DNA | TTCACTTCCCCTTGGCCACAACAT | TGTTCCATGCAGGGGAAAACAAGC |

## NAD$^+$/NADH quantification

For NAD$^+$/NADH quantifications, $5 \times 10^5$ cells were seeded in 10-cm cell culture dishes for technical duplicates and incubated for 48 hr. NAD$^+$/NADH ratios were quantified using a fluorometric kit assay (#MAK460-1KT, Sigma-Aldrich) following the manufacturer's instructions. For sample preparation, $9 \times 10^5$ cells were pelleted and homogenized with corresponding extraction buffers to be in the linear range of the kit. Fluorescence intensities were measured at $\lambda_{Ex}$ = 530 nm/$\lambda_{Em}$ = 585 nm on SPARK 10 M (TECAN) and data were exported to Prism (10.2.2, GraphPad) software to generate final figures.

## Seahorse experiments

Seahorse XFe96 cell culture microplates (Agilent) were coated with 50 μg/ml of poly-D-lysine (#A3890401, Thermo Fisher Scientific) in D-PBS (#311-425-CL, Wisent) for 1 hr at room temperature and extensively washed with sterile water. Seahorse XFe96 sensor cartridges (Agilent) were calibrated according to the manufacturer's protocol. Assay media were prepared as follows: 828 g DMEM without L-glutamine, phenol red, sodium pyruvate, and sodium bicarbonate (#219-060-XK, Wisent), 29.2 mg L-glutamine (#AC386032500, Fisher Chemical), 450.4 mg D-glucose (#D-16-500, Fisher Chemical), 100 μL HEPES 1 M (#330-050, Wisent), completed at 100 ml with Milli-Q water and adjusted to pH 7.2. Cells were plated

in poly-D-lysine-coated microplates at a density of 40,000 cells/well in 100 µl of assay media. Following a 30-min incubation at room temperature, 75 µl of assay media was added to each well, and the plate was transferred to an XFe96 analyzer. Typical runs were composed of the following steps: (1) baseline measurements for 18 min (3 cycles, each of 3-min mixing and 3-min measurement), (2) single drug injection (5 mM metformin, 100 µM phenformin, or sterile water as vehicle control), and (3) measurements over a period of 10.5 hr (63 cycles, each of 6-min mixing, 1-min pause, and 3-min measurement). OCRs and ECARs were extracted from the Wave software (Agilent) and all time points were normalized to the first measurement. Data were then exported to Prism (10.2.2, GraphPad) software to generate final figures.

## Seahorse Bioenergetic Mito Stress Test

Cells were seeded in poly-D-lysine-coated XF96 microplates at a density of 40,000 cells per well in 100 µl of complete culture medium (see *Cell culture* in *Materials and methods*). After a 1-hr incubation at room temperature to allow cell attachment, the medium was replaced with Seahorse assay medium described above by performing three gentle washing steps, leaving 50 µl per well. Cells were then treated with 125 µl of the appropriate condition: either metformin (5 mM), phenformin (100 µM), or sterile water as vehicle control. Plates were incubated for 3 hr at 37°C in a non-$CO_2$ incubator before being transferred to the Seahorse XFe96 analyzer. Typical runs consisted of a baseline measurement phase followed by sequential injections of mitochondrial inhibitors: oligomycin (10 µM), FCCP (3 µM), and a mix of rotenone and antimycin A (1 µM each). Each injection step included three measurement cycles (3-min mixing and 3-min measurement), for a total of 18 min per compound. OCRs and ECARs were obtained using the Wave software (Agilent).

## Crystal violet cell number assay

The crystal violet staining retention assay (*Chen et al., 2016*) was used to estimate cell growth and viability by measuring absorbance at 590 nm in a microplate reader (SpectraMax 190 microplates, Molecular Devices). For the growth assay, 5000 cells were seeded in four different 6-well cell culture dishes in technical triplicates and incubated for 4 different time points up to 6 days. Of note, media were replaced every 48 hr. At indicated times, cells were washed twice in PBS and fixed in 1% glutaraldehyde solution (PBS) for 10 min at rt. After fixation, cells were washed twice in PBS. Once all time points were collected, plates were stained in a 0.05% crystal violet solution (PBS) for 30 min with agitation. After staining, plates were washed five times in water and were dried for 24 hr. The next day, crystal violet retained in cells was solubilized in 10% acetic acid solution for 15 min with agitation and transferred into a 96-well plate for absorbance measurement. Data were then exported to Prism (10.2.2, GraphPad) software to generate final figures.

For the cell viability assay, 1000 cells were seeded in 96-well plates in technical triplicates. After 24 hr, cells were treated at different concentrations with the corresponding drugs and were grown for 72 hr. Crystal violet staining was performed as described for the growth assay. The half minimal effective concentrations were obtained using a fit of a curve with nonlinear regression log(inhibitor) vs. response -variable slope (four parameters) from the corresponding dose response curves with Prism (10.2.2, GraphPad) software. Importantly, because growth rate was affected in ATP5I KO cells generating variabilities, cell viability assays comparing ATP5I KO with control cells were done in media supplemented with pyruvate and uridine (see generation of stable ATP5I KO cells section). Of note, drugs were dissolved in the corresponding media.

## ATP quantification

For ATP measurements, $3 \times 10^3$ cells were seeded per well in a 96-well plate in technical duplicates. ATP levels (in picomoles) were determined using the ATP Determination Kit (#A22066, Invitrogen), according to the manufacturer's instructions. Prior to incubation, cells were washed three times with DMEM without phenol red (#319-080 CL, Wisent). After a 15-min incubation with firefly luciferase, luminescence was recorded at an emission wavelength of 560 nm ($\lambda_{em}$ = 560 nm) using a SPARK 10 M plate reader (TECAN). The data were subsequently exported to GraphPad Prism (v10.2.2) for figure generation.

## Chemogenomic CRISPR/Cas9 KO screen in NALM-6 cells

The genome-wide pooled CRISPR/Cas9 KO screens made in the presence of chemicals inhibiting growth were performed by the ChemoGenix platform (IRIC, Université de Montréal; https://chemogenix.iric.

ca/) as previously described (*Bertomeu et al., 2018*). Briefly, a NALM-6 clone bearing an integrated doxycycline-inducible Cas9 expression cassette generated by lentiviruses made from pCW-Cas9 (Addgene #50661) was transduced with the genome-wide KO EKO sgRNA library (*Bertomeu et al., 2018*) (278,754 different sgRNAs). After thawing the library from liquid $N_2$ and letting it recover in 10% FBS RPMI for 1 day, KOs were induced for 7 days of culture with 2 µg/ml doxycycline. The pooled library was then split into different T-75 flasks ($28 \times 10^6$ cells per flask; a representation of 100 cells/sgRNA) in 70 ml at $4 \times 10^5$ cells/ml. Cells were treated with 16 mM Metformin (using 1 M stock solution prepared in media) or 70 nM rotenone (Sigma, R8875) or 2 µM oligomycin A (Tocris Bioscience, 4110) (both from 1000X stock solutions prepared in DMSO) for 8 days with monitoring of growth every 2 days, diluting back to $4 \times 10^5$ cells/ml and adding more compounds to maintain the same final concentration whenever cells reached $8 \times 10^5$ cells/ml. Over that period, treated cells had respectively 1.47, 2.93, and 4.14 population doublings whereas no solvent or DMSO controls had about 7.5. Cells were collected, genomic DNA extracted using the Gentra Puregene kit according to the manufacturer's instructions (QIAGEN), and sgRNA sequences PCR-amplified as described (*Bertomeu et al., 2018*). SgRNA frequencies were obtained by next-generation sequencing (Illumina NextSeq 500). Reads were aligned using Bowtie2.2.5 in the forward direction only (norc option) with otherwise default parameters and total read counts per sgRNA tabulated. Context-dependent chemogenomic interaction scores were calculated from comparing sgRNA frequency changes against negative controls from using a modified version of the RANKS algorithm (*Bertomeu et al., 2018*) which uses guides targeting similarly essential genes as controls to distinguish condition-specific chemogenomic interactions from nonspecific fitness/essentiality phenotypes. Raw read counts are available upon request directly from the ChemoGenix platform.

## Statistical analysis

Statistical analyses were performed using GraphPad Prism (10.2.2) software. Unpaired two-tailed Student's *t*-test or Ordinary one-way ANOVA with compensation of multiple comparisons with Sidak's test were used to determine significance except for Seahorse analysis where paired two-tailed Student's *t*-test and RM one-way ANOVA were preferred. A value of $p < 0.05$ was considered significant. All specific statistical details can be found in the corresponding legends.

## Acknowledgements

We wish to express our deepest gratitude to IRIC proteomics and genomics platformes, Professor Omichinski for his expert guidance and unwavering support in the purification of proteins, and also to Professor Masson and Affinité Instrument for their generosity in providing access to the SPR system. SPG is supported by Grants 840633, 1054571 from The Cancer Research Society. ARS by Cancer Research Society (CRS) and Charlotte Légaré Memorial Fund 935858 and NSERC RGPIN-2021-03128. GF by Grants from the Terry Fox Research Institute and The Cancer Research Society and the CIBC chair for breast cancer research.

## Additional information

### Funding

| Funder | Grant reference number | Author |
|---|---|---|
| Terry Fox Research Institute | TFRI Project #1123 | Gerardo Ferbeyre |
| Cancer Research Society | Operating Grant 2016 | Gerardo Ferbeyre |
| Cancer Research Society | 935858 | Andreea R Schmitzer |
| Cancer Research Society | 840633 | Simon-Pierre Gravel |
| Natural Sciences and Engineering Research Council of Canada | RGPIN-2021-03128 | Andreea R Schmitzer |

| Funder | Grant reference number | Author |
|---|---|---|
| Cancer Research Society | Ganotec/Marc-André Pigeon Fund | Gerardo Ferbeyre |
| Cancer Research Society | 1054571 | Simon-Pierre Gravel |

The funders had no role in study design, data collection, and interpretation, or the decision to submit the work for publication.

## Author contributions

Guillaume Lefrançois, Conceptualization, Data curation, Formal analysis, Investigation, Methodology, Writing – original draft, Writing – review and editing; Emilie Lavallée, Conceptualization, Investigation, Methodology; Marie-Camille Rowell, Conceptualization, Formal analysis, Investigation, Methodology; Véronique Bourdeau, Data curation, Formal analysis, Investigation, Methodology; Farzaneh Mohebali, Thierry Bertomeu, Ana Maria Duman, Maya Nikolova, Investigation, Methodology; Mike Tyers, Methodology; Simon-Pierre Gravel, Conceptualization, Data curation, Formal analysis, Supervision, Investigation, Methodology, Writing – original draft, Project administration, Writing – review and editing, Funding acquisition; Andreea R Schmitzer, Conceptualization, Resources, Data curation, Formal analysis, Funding acquisition, Validation, Investigation, Visualization, Methodology, Writing – original draft, Project administration, Writing – review and editing; Gerardo Ferbeyre, Conceptualization, Resources, Data curation, Formal analysis, Supervision, Funding acquisition, Validation, Investigation, Visualization, Methodology, Writing – original draft, Project administration, Writing – review and editing

## Author ORCIDs

Véronique Bourdeau (ID) https://orcid.org/0000-0002-9044-2106
Thierry Bertomeu (ID) https://orcid.org/0000-0002-5313-7057
Simon-Pierre Gravel (ID) https://orcid.org/0000-0001-8411-8054
Andreea R Schmitzer (ID) https://orcid.org/0000-0002-3806-9076
Gerardo Ferbeyre (ID) https://orcid.org/0000-0002-2146-618X

Reviewer #1 (Public review): https://doi.org/10.7554/eLife.102680.3.sa1
Reviewer #2 (Public review): https://doi.org/10.7554/eLife.102680.3.sa2
Reviewer #3 (Public review): https://doi.org/10.7554/eLife.102680.3.sa3
Author response https://doi.org/10.7554/eLife.102680.3.sa4

# Additional files

## Supplementary files

MDAR checklist

## Data availability

All data generated in this study are included in the manuscript and supporting files; source data files have been provided for all figures.

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
