## [Editor Report · eLife Assessment]

This **valuable** manuscript describes ATP5I, a subunit of F1Fo-ATP synthase, as a key target of medicinal biguanides. The knockout of ATP5I in pancreatic cancer cells mimics biguanide treatment, inducing a metabolic switch from OXPHOS to glycolysis due to a compromised expression of the Complex I protein NDUFB8. This results in a markedly decreased NAD/NADH ratio and decreased cell proliferation. These **solid** findings point out ATP5I as a promising mitochondrial target for cancer therapies and contribute to our understanding of metformin's mechanism of action since many of its molecular mechanisms remain poorly understood.

---

## [Referee Report · Reviewer #1 (Public review)]

Summary:

In the manuscript entitled 'The Role of ATP Synthase Subunit e (ATP5I) in 1 Mediating the Metabolic and Antiproliferative 2 Effects of Biguanides', Lefrancois G et al. identifies ATP5I, a subunit of F1Fo-ATP synthase, as a key target of medicinal biguanides. ATP5I stabilizes F1Fo-ATP synthase dimers, essential for cristae morphology, but its role in cancer metabolism is understudied. The research shows ATP5I interacts with a biguanide analogue, and its knockout in pancreatic cancer cells mimics biguanide treatment effects, including altered mitochondria, reduced OXPHOS, and increased glycolysis. ATP5I knockout cells resist biguanide-induced antiproliferative effects, but reintroducing ATP5I restores the effects of metformin and phenformin. These findings highlight ATP5I as a promising mitochondrial target for cancer therapies. The manuscript is well written.

Strengths:

Demonstrated the experiments in a systematic and well accepted methods

Weaknesses:

Significance of the target molecule and mechanisms may help in understanding the molecular mechanisms of metformin.

Comments on revisions:

In the revised manuscript, the authors addressed all the queries.

---

## [Referee Report · Reviewer #2 (Public review)]

Summary:

The mechanism(s) by which the therapeutic drug metformin lowers blood glucose in type 2 diabetes and inhibits cell proliferation at higher concentrations remain contentious. Inhibition of complex 1 of the mitochondrial respiratory chain with consequent changes in cellular metabolites which favour allosteric activation of phosphofructokinase-1, allosteric inhibition of fructose bisphosphatase-1 and cAMP signalling and activation of AMPK which phosphorylates transcription factors are candidate mechanisms. The current manuscript proposes the e-subunit of ATP-synthase as a putative binding protein of biguanides and demonstrates that it regulates the expressivity of the Complex 1 protein NDUFB8.

Strengths:

(1) The metformin conjugate and metformin show comparable efficacy on inhibition of cell proliferation in the millimolar range.

(2) Demonstration of compromised expression of the Complex I protein NDUFB8 by the ATP5I knock out and its reversal by ATP5I expression is an important strength of the study. This shows that the decreased "sensitivity" to metformin in the ATP5I knock out cells could be due to various proteins.

(3) Demonstration of converse effects of ATP5I KO and re-expression ATP5I on the NAD/NADH ratio.

Weaknesses:

(1) The interpretation of the cellular co-localization of the biotin-biguanide conjugate with TOMM20 (Figure 1-D) as mitochondrial "accumulation" of the conjugate is overstated because it cannot exclude binding of the conjugate to the mitochondrial membrane. It would have been more convincing if additional incubations with the biotin-biguanide conjugate in combination with metformin had shown that metformin is competitive with the biotin-conjugate.

(2) The manuscript reports the identification of 69 proteins by mass spectrometry of the pull-down assay of which 31 proteins were eluted by metformin. However, no Mass Spectrometry data is presented of the peptides identified. The methodology does not state the minimum number of peptides (1, 2?) that were used for the identification of the 31/69 proteins.

(3) The validation of ATP5I was based on the use of recombinant protein (which was 90% pure) for the SPR and use of a single antibody to ATP5I. The validity of the immunoblotting rests on the assumption that there is no "non-specific" immunoactivity in the relevant mol wt range. Information on the validation of the antibody would be helpful.

(4) Knock-out of ATP5I markedly compromised the NAD/NADH ratio (Fig.3A) and cell proliferation (Fig.3D). These effects may be associated with decreased mitochondrial membrane potential which could explain the low efficacy for metformin (and most of the data in Figs 3-5). This possibility should be discussed. Effects of [metformin] on the NAD/NADH ratio in control cells and ATP5I-KO would have been helpful because the metformin data on cell growth is normalized as fold change relative to control, whereas the NAD/NADH ratio would represent a direct absolute measurement enabling comparison of the absolute effect in control cells with ATP5I KO.

(5) Figure-6 CRISPR/Cas9 KO at 16mM metformin in comparison with 70nM rotenone and 2 micromolar oligomycin (in serum containing medium). The rationale for use of such a high concentration of metformin has not been explained. In liver cells metformin concentrations above 1mM cause severe ATP depletion, whereas therapeutic (micromolar) concentrations have minimal effects on cellular ATP status. The 16mM concentration is ~2 orders of magnitude higher than therapeutic concentrations and likely linked to compromised energy status. The stronger inhibition of cell proliferation by 16mM metformin compared with rotenone or oligomycin raises the issue whether the changes in gene expression may be linked to the greater inhibition of mitochondrial metabolism. Validation of the cellular ATP status and NAD/NADH with metformin as compared with the two inhibitors could help the interpretation of this data.

Comments on revisions:

No further comments.

---

## [Referee Report · Reviewer #3 (Public review)]

Most of the data are based on measurements of the oxygen consumption rate (OCR) and extracellular acidification rate (ECAR) measured by the Seahorse analyser in control and ATP5l KO cells. However, these measurements are conducted by a single injection of a biguanide, followed over time and presented as fold change. By doing so, the individual information of the effect to of metformin and derivate on control and KO cells are lost. In addition, the usual measurement of OCR is coupled with certain inhibitors and uncouplers, such as oligomycin, FCCP and Antimycin A/rotenone, to understand the contribution of individual complexes to the respiration. Since biguanides and ATP5l KO affect protein levels of components of complex I and IV, it would be informative to measure their individual contributions/effects in the Seahorse. To further strengthen the data, it would be helpful to obtain measurements of actual ATP levels in these cells, as this would explain the activation of AMPK.

The authors report on alterations in mitochondrial morphology upon ATP5l KO, which is measured by subjective quantifications of filamentous versus puncta structures. Fiji offers great tools to quantify the mitochondrial network unbiased and with more accuracy using deconvolution and skeletonization of the mitochondria, providing the opportunity to measure length, shape and number quantitatively. This will help to understand better, whether mitochondria are really fragmented upon ATP5l KO and rescued by its re-introduction.

Finally, the authors report in the last part of the paper a genetic CRISPR/Cas9 KO screen in NALM-6 cells cultured with high amounts of metformin to identify potential new mediators of metformin action. It is difficult to connect that to the rest of the paper, because (a) different concentrations of metformin are used and (b) the metabolic effects on energy consumption are not defined. They argue about molecular function of the obtained hits based on literature, and on comparison the pattern of genetic alterations based on treatments with known inhibitors such as oligomycin and rotenone. However, a direct connection is not provided, thus the interpretation at the end of the results that "the OMA1-DEL1-HRI pathway mediates the antiproliferative activity of both biguanides and the F1ATPase inhibitor oligomycin" while increasing glycolysis, needs to be tuned down. This is an interesting observation, but no causality is provided. In general, this part stands alone and needs to be better connected to the rest of the paper.

Comments on revisions:

Thanks to the authors for addressing the concerns raised during the review of the original manuscript. The data now include proper measurements of OCR and quantifications of the mitochondria network. The screening data is better connected to the rest of the paper and provide compelling evidence for mitochondria and in particular the ATP synthase as potential targets of metformin.

---

## [Author Response]

The following is the authors’ response to the original reviews.

**Public Reviews:**

**Reviewer #1 (Public review):**
Summary:In the manuscript entitled 'The Role of ATP Synthase Subunit e (ATP5I) in 1 Mediating the Metabolic and Antiproliferative 2 Effects of Biguanides', Lefrancois G et al. identifies ATP5I, a subunit of F1Fo-ATP synthase, as a key target of medicinal biguanides. ATP5I stabilizes F1Fo-ATP synthase dimers, essential for cristae morphology, but its role in cancer metabolism is understudied. The research shows ATP5I interacts with a biguanide analogue, and its knockout in pancreatic cancer cells mimics biguanide treatment effects, including altered mitochondria, reduced OXPHOS, and increased glycolysis. ATP5I knockout cells resist biguanide-induced antiproliferative effects, but reintroducing ATP5I restores the effects of metformin and phenformin. These findings highlight ATP5I as a promising mitochondrial target for cancer therapies. The manuscript is well written.Strengths:Demonstrated the experiments in systematic and well-accepted methods.Weaknesses:The significance of the target molecule and mechanisms may help in understanding the molecular mechanisms of metformin.

We greatly appreciate the reviewer’s insightful comment regarding the importance of the target molecule and its mechanisms in elucidating metformin’s molecular actions. ATP5I plays a key role in the dimerization and assembly of the F1F0-ATP synthase complex. To address this, we performed Blue Native-PAGE followed by western blotting using an antibody against the β-subunit of the F1 domain. Our results show that metformin affects the oligomeric state of the F1F0-ATP synthase in a way that partially reproduces the effect of the KO of ATP5I (Fig 2G). This provides direct evidence that metformin acts on-target through ATP5I.

**Reviewer #2 (Public review):**
Summary:The mechanism(s) by which the therapeutic drug metformin lowers blood glucose in type 2 diabetes and inhibits cell proliferation at higher concentrations remain contentious. Inhibition of complex 1 of the mitochondrial respiratory chain with consequent changes in cellular metabolites which favour allosteric activation of phosphofructokinase-1, allosteric inhibition of fructose bisphosphatase-1 and cAMP signalling and activation of AMPK which phosphorylates transcription factors are candidate mechanisms. The current manuscript proposes the e-subunit of ATP-synthase as a putative binding protein of biguanides and demonstrates that it regulates the expressivity of the Complex 1 protein NDUFB8.Strengths:(1) The metformin conjugate and metformin show comparable efficacy on inhibition of cell proliferation in the millimolar range.(2) Demonstration of compromised expression of the Complex I protein NDUFB8 by the ATP5I knockout and its reversal by ATP5I expression is an important strength of the study. This shows that the decreased "sensitivity" to metformin in the ATP5I knock-out cells could be due to various proteins.(3) Demonstration of converse effects of ATP5I KO and re-expression ATP5I on the NAD/NADH ratio.Weaknesses:(1) The interpretation of the cellular co-localization of the biotin-biguanide conjugate with TOMM20 (Figure 1-D) as mitochondrial "accumulation" of the conjugate is overstated because it cannot exclude binding of the conjugate to the mitochondrial membrane. It would have been more convincing if additional incubations with the biotin-biguanide conjugate in combination with metformin had shown that metformin is competitive with the biotin-conjugate.

We appreciate the reviewer’s comment and agree that the resolution provided by fluorescence microscopy makes it challenging to pinpoint the specific mitochondrial compartment where the biotin-biguanide conjugate localizes, even with additional markers such as TOMM20 antibodies for the inner mitochondrial membrane. While it remains a possibility that the conjugate binds to the mitochondrial surface, another plausible explanation is that the biotin moiety may facilitate entry into mitochondria through a biotin-specific transporter, adding further mechanistic intricacies. Furthermore, while a competition assay with metformin might help investigate interactions with mitochondrial targets and transporters (OCT family), it would not compete for biotin-mediated transport. Thus, while we acknowledge the reviewer’s suggestion, we believe such an experiment may not provide conclusive evidence regarding the conjugate’s mitochondrial localization or mechanism of entry. Instead, we revised the manuscript to more accurately describe the findings as "mitochondrial association" rather than "mitochondrial accumulation," ensuring that our interpretation remains consistent with the resolution and limitations of the data presented.

(2) The manuscript reports the identification of 69 proteins by mass spectrometry of the pull-down assay of which 30 proteins were eluted by metformin. However, no Mass Spectrometry data is presented of the peptides identified. The methodology does not state the minimum number of peptides (1, 2?) that were used for the identification of the 31/69 proteins.

We added a comprehensive table summarizing these findings (Figure 1- figure supplement 2). We considered all peptides and decided to perform stringent validation tests for those chosen to be further studied.

(3) The validation of ATP5I was based on the use of recombinant protein (which was 90% pure) for the SPR and the use of a single antibody to ATP5I. The validity of the immunoblotting rests on the assumption that there is no "non-specific" immunoactivity in the relevant mol wt range. Information on the validation of the antibody would be helpful.

Regarding the recombinant protein used for SPR, its purity was evaluated using a Coomassie-stained gel. For the antibody used in immunoblotting, its specificity was validated through knockout cell lines (Figure 2A), ensuring minimal concerns about non-specific immunoactivity within the relevant molecular weight range. Unfortunately, the KO data comes in the paper after the first immunoblots are presented. We outlined this validation in the methods section.

(4) Knock-out of ATP5I markedly compromised the NAD/NADH ratio (Fig.3A) and cell proliferation (Figure 3D). These effects may be associated with decreased mitochondrial membrane potential which could explain the low efficacy of metformin (and most of the data in Figures 3-5). This possibility should be discussed. Effects of [metformin] on the NAD/NADH ratio in control cells and ATP5I-KO would have been helpful because the metformin data on cell growth is normalized as fold change relative to control, whereas the NAD/NADH ratio would represent a direct absolute measurement enabling comparison of the absolute effect in control cells with ATP5I KO.

The mitochondrial membrane potential depends on a functional electron transport chain which drives proton pumping from the matrix to the intermembrane space. Metformin can decrease the mitochondrial membrane potential and this is usually explained as a consequence of complex I inhibition [1]. It has been published that metformin requires this membrane potential to accumulate in mitochondria so the actions of metformin are self-limiting due to this requirement. The reviewer is right that ATP5I KO cells could be resistant to metformin because they may have a lower membrane potential. We do not believe this to be the case because the response to phenformin, another biguanide that can enter mitochondria through the membrane without the need of the OCT transporters [2], is also affected in ATP5I KO cells. Of note, compensatory mechanisms such as enhanced glycolysis, as observed in ATP5I KO cells (elevated ECAR and increased sensitivity to 2-d-deoxyglucose), and the ATPase activity of F_1_F_0_-ATP synthase could potentially help maintain membrane potential suggesting that this might not be an issue in the ATP5I KO cells. Chandel and colleagues already proposed that reversal of the F_1_F_0_-ATPase keeps this membrane potential in metformin-treated cells [3].

Nevertheless, to experimentally address this point, we measured the mitochondrial membrane potential using tetramethylrhodamine methyl ester (TMRE) and ATP levels using luciferase-based assays (CellTiter-Glo) in ATP5I KO cells. We sow now that ATP levels are not significantly reduced in ATP5I KO cells, likely because of compensatory glycolysis (Figure 5D), while the mitochondrial membrane potential remains close to normal (Figure 6D and E).

We did not measure the NAD^+^/NADH in both control and KO cells treated with metformin because we provide now a more direct measurement of metformin acting on ATP5I: the state of oligomerization of the F_1_F_0_-ATPase (Figure 2G) as well as a Seahorse Bioenergetic Stress test (Figure 6A-C). Both figures provide results consistent with targeting ATP5I by biguanides. We also discuss that targeting ATP5I can result in complex I inhibition due to the well-known role of F_1_F_0_-ATPases in cristae formation and the assembly of the respiratory complexes. We do not believe ATP5I is the only target of metformin and in the paper we properly acknowledged and discussed other proposed targets in the introduction, results section page 8 and the discussion.

(5) Figure-6 CRISPR/Cas9 KO at 16mM metformin in comparison with 70nM rotenone and 2 micromolar oligomycin (in serum-containing medium). The rationale for the use of such a high concentration of metformin has not been explained. In liver cells metformin concentrations above 1mM cause severe ATP depletion, whereas therapeutic (micromolar) concentrations have minimal effects on cellular ATP status. The 16mM concentration is ~2 orders of magnitude higher than therapeutic concentrations and likely linked to compromised energy status. The stronger inhibition of cell proliferation by 16mM metformin compared with rotenone or oligomycin raises the issue of whether the changes in gene expression may be linked to the greater inhibition of mitochondrial metabolism. Validation of the cellular ATP status and NAD/NADH with metformin as compared with the two inhibitors could help the interpretation of this data.

NALM-6 cells are very glycolytic, have low respiration rates, and weak dependence on ATP5I (DepMap score: -0.47) [4]. The concentration of 16 mM metformin was chosen based on the IC_50_ for this cell line. Both ATP status and NAD^+^/NADH ratios will depend on the extent of the compensatory glycolysis. On the other hand, our genetic screening evaluates cell proliferation as an integration of all metabolic activities required for the process. This unbiased screening revealed a common pathway affected by metformin and oligomycin different that the pathway affected by rotenone, which is consistent with the finding that metformin acts of the F_1_F_0_-ATPase. Our new Seahorse data demonstrate that oligomycin has a markedly reduced effect in metformin-treated cells, supporting a shared mechanism of action. Notably, uncouplers restore respiration in both metformin-treated and ATP5I knockout cells, which aligns with the mechanism we propose (please see our new section on the Seahorse Mito Stress test and the new discussion). In the discussion, we acknowledged—based on existing literature—that the cellular context may play a significant role in determining the response to this drug.

**Reviewer #3 (Public review):**
Most of the data are based on measurements of the oxygen consumption rate (OCR) and extracellular acidification rate (ECAR) measured by the Seahorse analyser in control and ATP5l KO cells. However, these measurements are conducted by a single injection of a biguanide, followed over time and presented as fold change. By doing so, the individual information on the effect of metformin and derivate on control and KO cells are lost. In addition, the usual measurement of OCR is coupled with certain inhibitors and uncouplers, such as oligomycin, FCCP, and Antimycin A/rotenone, to understand the contribution of individual complexes to respiration. Since biguanides and ATP5l KO affect protein levels of components of complex I and IV, it would be informative to measure their individual contributions/effects in the Seahorse. To further strengthen the data, it would be helpful to obtain measurements of actual ATP levels in these cells, as this would explain the activation of AMPK.

Thank you for this valuable comment. We have now performed the suggested analysis, which is presented in the new Figure 6. The data are consistent with our proposition that biguanides target ATP5I, but they also suggest the possibility of additional targets, such as Complex I, as proposed by other groups. Please see our new section on the Seahorse Mito Stress test and the new discussion. We also measured ATP (Figure 5D). and the mitochondrial membrane potential (Figure 6D and E). These measurements reflect the powerful compensation provided by glycolysis.

The authors report on alterations in mitochondrial morphology upon ATP5l KO, which is measured by subjective quantifications of filamentous versus puncta structures. Fiji offers great tools to quantify the mitochondrial network unbiasedly and with more accuracy using deconvolution and skeletonization of the mitochondria, providing the opportunity to measure length, shape, and number quantitatively. This will help to understand better, whether mitochondria are really fragmented upon ATP5l KO and rescued by its re-introduction.

Thanks for the suggestion. We used the Mitochondrial analyzer plugin from ImageJ/Fiji and redid Figure 2 and 4 and quantified details of the mitochondrial network reporting differences in branches number, length, endpoints and diameter.

Finally, the authors report in the last part of the paper a genetic CRISPR/Cas9 KO screen in NALM-6 cells cultured with high amounts of metformin to identify potential new mediators of metformin action. It is difficult to connect that to the rest of the paper because (a) different concentrations of metformin are used and (b) the metabolic effects on energy consumption are not defined. They argue about the molecular function of the obtained hits based on literature and on a comparison of the pattern of genetic alterations based on treatments with known inhibitors such as oligomycin and rotenone. However, a direct connection is not provided, thus the interpretation at the end of the results that "the OMA1-DEL1-HRI pathway mediates the antiproliferative activity of both biguanides and the F1ATPase inhibitor oligomycin" while increasing glycolysis, needs to be toned down. This is an interesting observation, but no causality is provided. In general, this part stands alone and needs to be better connected to the rest of the paper.

NALM-6 are very glycolytic, have low respiration rates, and weak dependence on ATP5I [4], forcing us to use higher concentrations of metformin to inhibit their growth. Recent results show that metformin targets PEN2 in the cytosol to increase AMPK activity, controlling both the glucose lowering and the life span extension abilities of metformin [5]. This work raises the question whether the antiproliferative and anticancer effects of metformin are due to a mitochondrial activity or are controlled by this new pathway of AMPK activation. Hence, the genetic screening was performed to unbiasedly find how metformin works. The results provide compelling evidence for mitochondria and in particular the ATP synthase as potential targets of metformin and a foundation for future studies. We added to the following text to the beginning of this section: “Several candidate targets have been reported for biguanides and our results presented so far suggest a new one. Clues about drug mechanism of action can be obtained in unbiased manner using genetic perturbation [6]. To obtain an unbiased observation of biological processes affected by metformin, we performed a genome-wide pooled CRISPR/Cas9 KO screen in NALM-6 cells cultured in the presence of metformin at a concentration affecting growth (16 mM).”

**Recommendations for the authors:**

**Reviewer #1 (Recommendations for the authors):**
(1) In Figure 1B, the total ACC antibody is missing, and the total AMPK should be replaced, especially since they claim pAMPK increases with metformin and BFB treatment. Additionally, the streptavidin pull-down image in Figure 1F needs to be resized to show the fully cropped section.

We repeated this experiment three times and added the new figures to the supplemental data. We corrected the main figure in the manuscript with a representative blot for total ACC (Fig 1B).

(2) Clarify whether ATP5I alone activates mitochondrial respiratory activity or if it functions in a complex with other proteins. Also, explain how metformin affects ATP5I-is it phosphorylated directly or through an upstream target

ATP5I interacts directly with ATP5L and both proteins form part of the peripheral stack of the F_1_F_0_-ATP synthase. ATP5I and ATP5L play demonstrated roles in the dimerization of the F_1_F_0_-ATP synthase. We discussed that they may affect other functions of the enzyme as part of the peripheral stack which interact with the OSCP (oligomycin sensitivity conferring protein) located in the F1 portion of the enzyme. Further work is needed to understand how ATP5I may affect the interactions between the F0 and F1 parts of the enzyme. We did not investigate whether metformin affects the phosphorylation of ATP5I, but this remains an important question for future studies. The PhosphoSitePlus database indicates that ATP5I undergoes phosphorylation and acetylation at multiple sites, suggesting potential regulatory mechanisms worth exploring.

(3) Ensure that all immunofluorescence (IF) images include a scale bar.

Done

**Reviewer #2 (Recommendations for the authors):**
(1) Details of the mass spectrometry analysis and the number of peptides for the proteins identified would increase the merit of the study.

We added a comprehensive table summarizing these findings (Figure 1- figure supplement 2). We considered all peptides and decided to perform stringent validation tests for those chosen to be further studied.

(2) The lower NAD/NADH ratios in the ATP5I KO cell lines and the higher ratios with ATP5I expression are convincing data of the cellular redox state of these cells (with variable NDUFB8). Other data sets (e.g. OCR and ECAR and Relative growth, %) are normalized to the respective control and therefore do not show the relative effect of metformin (in control cells) to the ATP5I knock-out. The effects of metformin concentration on the NAD/NADH ratio would provide a direct measure of the extent to which metformin mimics ATP5I KO. This data would be clearer to interpret than Figure 3GHKL; Figures 5EF; S1; S2.

We did not measure the NAD^+^/NADH in both control and KO cells treated with metformin because we provide now a more direct measurement of metformin acting on ATP5I: oligomerization state F_1_F_0_-ATPase and its vestigial assembly intermediates (Figure 2G) as well as a Seahorse Bionergetic Stress test (Figure 6A-C). Both figures provide results consistent with targeting ATP5I by biguanides. We also discuss that targeting ATP5I can result in complex I inhibition due to the well-known role of F_1_F_0_-ATPases oligomerization in cristae formation and the assembly of the respiratory complexes.

(3) Figure 6: NAD/NADH data for metformin (16mM) and rotenone (70 nM) /oligomycin (2 uM) would establish whether the concentrations are "matched" to allow a comparison of their gene signatures.

We used those concentrations based on similar effects on cell growth since the ration NAD/NADH depends on the extent of glycolytic compensation induced by blocking respiration.

(4) Intramitochondrial accumulation of the biotin conjugate could be demonstrated in Figure 1D from competition between metformin and the biotin-conjugate.

We appreciate the reviewer’s comment and agree that the resolution provided by fluorescence microscopy makes it challenging to pinpoint the specific mitochondrial compartment where the biotin-biguanide conjugate localizes, even with additional markers such as TOMM20 antibodies for the inner mitochondrial membrane. While it remains a possibility that the conjugate binds to the mitochondrial surface, another plausible explanation is that the biotin moiety may facilitate entry into mitochondria through a biotin-specific transporter, adding further mechanistic intricacies. Furthermore, while a competition assay with metformin might help investigate interactions with mitochondrial targets and transporters (OCT family), it would not compete for biotin-mediated transport. Thus, while we acknowledge the reviewer’s suggestion, we believe such an experiment may not provide conclusive evidence regarding the conjugate’s mitochondrial localization or mechanism of entry. Instead, we revised the manuscript to more accurately describe the findings as "mitochondrial association" rather than "mitochondrial accumulation," ensuring that our interpretation remains consistent with the resolution and limitations of the data presented.

**Reviewer #3 (Recommendations for the authors):**
In addition to my comments for the public review, the manuscript would be strengthened by the following points:(1) The abstract needs to be streamlined to communicate more clearly what the paper is about. The last part of the results is not mentioned and is completely disconnected from the ATP5I KO story.

We have significantly modified our abstract to include both the genetic screening significance and our new findings on the F_1_F_0_-ATP synthase oligomerization.

(2) Quantifications of the western blots (Figure 1B) are missing. Seems like AMPK total protein levels go down with BFB.

We quantified the blots.

(3) How often was the pull-down repeated (Figure 1F)? It would be also important to show this in other cell types, such as pancreatic cancer cells.

The pull-down was an initial large-scale discovery experiment performed once. However, the findings were subsequently validated in KP-4 pancreatic cancer cells in three independent experiments. As a direct readout of metformin’s impact on ATP5I, we assessed the oligomerization state of the F1ATPase and compared the effects of metformin with those of ATP5I knockout. We show that metformin partially phenocopies the ATP5I KO phenotype, and we reproduced this effect in a second cell line, U2OS osteosarcoma cells.

(4) Does the KO of ATP5l affect other subunits of the v-ATP5a?

Yes—we added an immunoblot to document this in Fig. 2A. Notably, ATP5I knockout also reduces ATP5L and OSCP levels.

(5) Does metformin and BFB itself affect mitochondrial morphology and respiration?

To evaluate the activity of BFB in comparison with metformin, we performed immunoblot analyses of the AMPK pathway, growth assays, and microscopy-based assessment of mitochondrial morphology. These data are shown in Fig. 1B–D. A more comprehensive analysis of metformin’s effects on mitochondrial respiration has now been added as Fig. 6, using Seahorse measurements and multiple respiratory inhibitors.

(6) Since there is a strong increase in ECAR, does this correspond to an increase in glucose uptake? Are the proteins or genes involved altered or how to explain the increased flux through glycolysis in ATP5l KO cells?

This is a very interesting idea, as our CRISPR screen identified several genes that could potentially enhance glycolysis as a vulnerability in metformin-treated cells. In future work, we will explore this biology in greater depth.

(7) Line 242, for easier understanding, states clearly that metformin reduces growth by x-percent.

Yes, is a 65-fold chang. We added it to the text.

(8) The conclusion at the end of the result section is not supported by the data or not well explained. I guess oligomycin will stop the action of metformin on vATP5l, or how to explain this?

We clarified the conclusion.

(1) Xian, H., Liu, Y., Rundberg Nilsson, A., Gatchalian, R., Crother, T. R., Tourtellotte, W. G., Zhang, Y., Aleman-Muench, G. R., Lewis, G., Chen, W., Kang, S., Luevanos, M., Trudler, D., Lipton, S. A., Soroosh, P., Teijaro, J., de la Torre, J. C., Arditi, M., Karin, M. & Sanchez-Lopez, E. Metformin inhibition of mitochondrial ATP and DNA synthesis abrogates NLRP3 inflammasome activation and pulmonary inflammation. Immunity 54, 1463-1477 e1411, (2021).

(2) Hawley, S. A., Ross, F. A., Chevtzoff, C., Green, K. A., Evans, A., Fogarty, S., Towler, M. C., Brown, L. J., Ogunbayo, O. A., Evans, A. M. & Hardie, D. G. Use of cells expressing gamma subunit variants to identify diverse mechanisms of AMPK activation. Cell metabolism 11, 554-565, (2010).

(3) Wheaton, W. W., Weinberg, S. E., Hamanaka, R. B., Soberanes, S., Sullivan, L. B., Anso, E., Glasauer, A., Dufour, E., Mutlu, G. M., Budigner, G. S. & Chandel, N. S. Metformin inhibits mitochondrial complex I of cancer cells to reduce tumorigenesis. eLife 3, e02242, (2014).

(4) Hlozkova, K., Pecinova, A., Alquezar-Artieda, N., Pajuelo-Reguera, D., Simcikova, M., Hovorkova, L., Rejlova, K., Zaliova, M., Mracek, T., Kolenova, A., Stary, J., Trka, J. & Starkova, J. Metabolic profile of leukemia cells influences treatment efficacy of L-asparaginase. BMC Cancer 20, 526, (2020).

(5) Ma, T., Tian, X., Zhang, B., Li, M., Wang, Y., Yang, C., Wu, J., Wei, X., Qu, Q., Yu, Y., Long, S., Feng, J. W., Li, C., Zhang, C., Xie, C., Wu, Y., Xu, Z., Chen, J., Yu, Y., Huang, X., He, Y., Yao, L., Zhang, L., Zhu, M., Wang, W., Wang, Z. C., Zhang, M., Bao, Y., Jia, W., Lin, S. Y., Ye, Z., Piao, H. L., Deng, X., Zhang, C. S. & Lin, S. C. Low-dose metformin targets the lysosomal AMPK pathway through PEN2. Nature 603, 159-165, (2022).

(6) Bruno, P. M., Liu, Y., Park, G. Y., Murai, J., Koch, C. E., Eisen, T. J., Pritchard, J. R., Pommier, Y., Lippard, S. J. & Hemann, M. T. A subset of platinum-containing chemotherapeutic agents kills cells by inducing ribosome biogenesis stress. Nat Med 23, 461-471, (2017).